# BRET-based RAS biosensors that show a novel small molecule is an inhibitor of RAS-effector protein-protein interactions

Nicolas Bery[1], Abimael Cruz-Migoni[1,2], Carole JR Bataille[3], Camilo E Quevedo[1], Hanna Tulmin[1†], Ami Miller[1], Angela Russell[3], Simon EV Phillips[4], Stephen B Carr[2,4], Terence H Rabbitts[1]*

[1]MRC Molecular Haematology Unit, Weatherall Institute of Molecular Medicine, University of Oxford, Oxford, United Kingdom; [2]Research Complex at Harwell, Rutherford Appleton Laboratory, Didcot, United Kingdom; [3]Chemistry Research Laboratory, Oxford, United Kingdom; [4]Department of Biochemistry, University of Oxford, Oxford, United Kingdom

**\*For correspondence:**
terence.rabbitts@imm.ox.ac.uk

**Present address:** [†]Wellcome Trust Centre for Human Genetics, Oxford, United Kingdom

**Abstract** The RAS family of proteins is amongst the most highly mutated in human cancers and has so far eluded drug therapy. Currently, much effort is being made to discover mutant RAS inhibitors and in vitro screening for RAS-binding drugs must be followed by cell-based assays. Here, we have developed a robust set of bioluminescence resonance energy transfer (BRET)-based RAS biosensors that enable monitoring of RAS-effector interaction inhibition in living cells. These include KRAS, HRAS and NRAS and a variety of different mutations that mirror those found in human cancers with the major RAS effectors such as CRAF, PI3K and RALGDS. We highlighted the utility of these RAS biosensors by showing a RAS-binding compound is a potent pan-RAS-effector interactions inhibitor in cells. The RAS biosensors represent a useful tool to investigate and characterize the potency of anti-RAS inhibitors in cells and more generally any RAS protein-protein interaction (PPI) in cells.
DOI: https://doi.org/10.7554/eLife.37122.001

## Introduction

*RAS* is the most prominent oncogene identified in cancer. Mutation in RAS proteins can be found in approximately 30% of all human tumors (*Downward, 2003*; *Prior et al., 2012*) (http://cancer.sanger.ac.uk/cosmic) prompting interest in the discovery of anti-RAS therapeutics. However, there are still no RAS-targeted drugs currently available in the clinic even though such molecules could prove widely efficacious in many human cancers as front-line drugs for therapy. Some forms of cancer, like pancreatic cancer, present late and are difficult therefore to treat (*Kleeff et al., 2016*) but these contain a high proportion of *KRAS* mutations and are thus potentially susceptible to RAS-binding drugs.

RAS has been regarded as undruggable partly because so far attempts to interfere with the protein have not been efficacious (*Cox et al., 2014*). RAS is a membrane-bound small GTPase switching between an inactive GDP-bound state and an active GTP-bound state. RAS signaling to the cell nucleus occurs after interaction of RAS-GTP with its effectors to trigger the activation of downstream signaling pathways. This activation thereby promotes cell survival and cell proliferation (*Wennerberg et al., 2005*) via gene modulation so that the blockade of mutant RAS signaling in tumors cells is an attractive therapeutic option. There are several ways in which this could be achieved (*Athuluri-Divakar et al., 2016*; *Burns et al., 2014*; *Spiegel et al., 2014*; *Zimmermann et al., 2013*) but methods such as implementing farnesylation inhibitors have limited

**eLife digest** A group of proteins known as the RAS family plays a critical role in controlling animal cell growth and division. RAS proteins are normally active only some of the time, but genetic mutations can create permanently active forms of the proteins. These constantly interact with other proteins called effectors. In response, cells multiply uncontrollably and give rise to cancers.

In an attempt to find new cancer treatments, researchers across the globe are trying to develop inhibitor drugs that prevent RAS and effector proteins from interacting. New drugs are often tested in laboratory experiments that directly apply the drugs to the proteins that they are designed to work on. But in some cases a drug may work wellin the laboratory but fail to work when used in cells. Unfortunately, there are few ways to judge how well inhibitor drugs work inside living cells.

Bery et al. have now developed RAS biosensors – a collection of proteins that bind to RAS and produce light more brightly when RAS interacts with effector proteins in living cells. Tests on cells treated with an antibody that works inside cells and is known to prevent interactions between RAS and effector proteins confirmed that the RAS biosensors work well. Bery et al. then used the RAS biosensors to show that a new RAS inhibitor works in human cancer cells.

The RAS biosensors are available upon request to researchers across the globe. They should form an important tool for testing potential treatments for cancers that contain mutated RAS proteins.

DOI: https://doi.org/10.7554/eLife.37122.002

success due to side effects (*Berndt et al., 2011*; *James et al., 1995*; *Whyte et al., 1997*). One avenue that has largely been avoided in inhibiting RAS is the interaction with its effectors, such as RAF, RALGDS and PI3K. However, the effectiveness of the orthosteric RAS-effector PPI inhibition was shown using intracellular antibodies (*Tanaka and Rabbitts, 2003*; *Tanaka et al., 2007*) (herein called macrodrugs (*Tanaka and Rabbitts, 2008*) to distinguish them from conventional small molecule drugs) and a single domain intracellular antibody that blocks effector interaction sites of RAS-GTP. This PPI inhibition can prevent tumor growth in xenograft models and tumor initiation in a transgenic mouse model (*Tanaka and Rabbitts, 2010*; *Tanaka et al., 2007*). Other macrodrugs, such as DARPins (*Guillard et al., 2017*), have also been shown to be effective in interfering with RAS PPIs. Moreover, for many years, RAS was regarded as a protein without any pockets suitable for small molecule interactions (*McCormick, 2016*) but recent studies have described compounds that are able to bind RAS-associated pockets (*Gentile et al., 2017*; *Lito et al., 2016*; *Maurer et al., 2012*; *Ostrem et al., 2013*; *Patricelli et al., 2016*; *Shima et al., 2013*; *Sun et al., 2012*; *Waldmann et al., 2004*; *Welsch et al., 2017*).

Most of the current RAS inhibitors have been selected and identified through in vitro techniques (*Ostrem et al., 2013*; *Trinh et al., 2016*; *Upadhyaya et al., 2015*; *Welsch et al., 2017*) but cell-based assay technologies are needed to assess initial hits for efficacy before hit to lead development is undertaken. Indeed, a robust cell-based assay is a mandatory step in any drug discovery programme, as it provides insights into the behavior of compounds in physiological conditions, including cell permeability, stability and potency in the cellular complexity of a whole cell. We now describe a toolbox of mutant and wild-type RAS BRET-based biosensors that can be used to assess PPI between activated, GTP-bound RAS (KRAS, HRAS or NRAS) and effectors such as CRAF, RALGDS or PI3K in living cells. We validate the toolbox using a published anti-RAS intracellular domain antibody (hereafter named iDAb RAS) (*Tanaka et al., 2007*), which is an inhibitor of RAS PPI to establish the RAS biosensor resource. We have further used this methodology to test a RAS-binding compound (herein referred to as 3344) that we have derived from an in vitro medicinal chemistry programme starting with an intracellular antibody fragment. By monitoring the change in BRET2-specific signal in transfected HEK293T cells expressing different RAS-effector donor-acceptor combinations, we have been able to characterize the pan-RAS-effector PPI inhibitor properties of 3344. This inhibitory mechanism shown using the BRET biosensor toolbox was supported by the crystal structure of KRAS with bound 3344, showing binding to a pocket close to the RAS switch. Therefore, the BRET2 toolbox we describe here is a critical resource and is available for all investigators in the international effort to produce anti-RAS drugs, that can be employed in the treatment of cancers with RAS mutations.

## Results

### Engineering and validation of mutant RAS biosensors

RAS biosensors were developed for use in the BRET2 method (*Bacart et al., 2008*) as a real-time system allowing the monitoring of protein-protein interactions and their inhibition in live cells. The scheme used is outlined in *Figure 1A*. The intracellular localization of BRET donor RAS proteins was recapitulated by expressing the full-length proteins including the CAAX box, which is the farnesylation site for trafficking to the plasma membrane. The CAAX sequences were fused to the carboxy terminal end of the *Renilla* Luciferase variant 8 (RLuc8) to act as the donor molecule in BRET2 (*De et al., 2007*) (for simplicity of the nomenclature, CAAX has been omitted from the RAS construct names). We used available structural data for RAS/effector and RAS/iDAb complexes to optimize the proximity of donor and acceptor moieties. Hence, RLuc8 was fused to the amino termini of full-length RAS family proteins and the GFP$^2$ (*Ramsay et al., 2002*) fused to the C-termini of the effectors (RALGDS, CRAF, PI3K) or of the iDAbs. Other parameters can influence the BRET2 signal such as the linker length between RLuc8/RAS and effector-iDAb/GFP$^2$. For our study, we observed a higher BRET signal with a (GGGS)$_3$ linker between RLuc8-KRAS$^{G12D}$ construct, a (GGGS)$_3$ linker between the CRAF RBD-GFP$^2$ molecule and a (GGGS)$_2$ linker between iDAb RAS-GFP$^2$ construct (*Figure 1—figure supplement 1A*). Therefore, we implemented these observations to all our BRET biosensors (*Supplementary file 1*). When donor and acceptor plasmids are transfected into HEK293T cells (although any cell line of choice would be suitable), the resultant cells are fluorescent and bioluminescent if treated with the luciferase substrate (coelenterazine 400a). If an interaction occurs between RAS and a partner-GFP$^2$ fusion, bringing the RLuc8 and GFP$^2$ within 100 Å, an energy transfer occurs from the RLuc8-RAS donor to the GFP$^2$ acceptor and a BRET2 signal is achieved (*Figure 1A*, middle panel). Inhibitors of the donor-acceptor molecule interaction will decrease the BRET signal whilst maintaining the RLuc8 bioluminescence and GFP$^2$ fluorescence signals (*Figure 1A*, right hand panel). The BRET signal (or BRET ratio) is calculated as the light emitted by the GFP$^2$ acceptor constructs (at 515 nm) upon addition of coelenterazine 400a, divided by the light emitted by the RLuc8 donor constructs (at 410 nm) (*Pfleger et al., 2006*). A background BRET signal is only observed with the donor-only construct where the RLuc8 plasmid is transfected alone into the cells (*Figure 1—figure supplement 1B*) and this signal is therefore subtracted from that BRET ratio. As shown in *Figure 1—figure supplement 1B*, un-transfected cells and those transfected with GFP$^2$-only construct have a negligible auto-luminescence and emission at 515 nm upon addition of the BRET substrate and are not considered in the calculation of the BRET ratio.

BRET donor saturation assessments were first carried out with the RAS effector RAS binding domains (RBDs) to evaluate the optimal levels of expression plasmid transfection for the competition experiments (*Figure 1B*). All of the effector domains were found to interact specifically with KRAS$^{G12D}$ since the BRET signal reached a donor saturation level (*Figure 1B*). Further, all the transfected plasmids expressed the proteins at equivalent levels as indicated by western blot analysis (*Figure 1C*) and their expression does not modify KRAS$^{G12D}$ expression (*Figure 1—figure supplement 2A* shows the increase of acceptor protein level has little effect of donor protein levels). To further characterize this BRET2 system, we used the dominant negative mutant KRAS$^{S17N}$, which does not interact with the effectors (*Cool et al., 1999*; *Nassar et al., 2010*; *van den Berghe et al., 1997*), as a donor. We found that the BRET signal increased linearly with the concentration of acceptor for all the RAS binding domains. This result is typical of non-specific interactions (*Mercier et al., 2002*), confirming the S17N mutant does not interact with the effectors and supports the sensitivity of this system (*Figure 1—figure supplement 2B*).

We initially characterized the biosensor pairs with the iDAb RAS that is known to interact with mutant KRAS on the switch regions (*Tanaka et al., 2007*), compared with a non-relevant anti-LMO2 iDAb (*Sewell et al., 2014*; *Tanaka et al., 2011*) that was designated as iDAb control in this study (herein called iDAb Ctl). Introduction of mutations in the three CDRs of the iDAb RAS to generate a dematured iDAb RAS (iDAb$_{dm}$ RAS), was shown to reduce its affinity towards RAS-GTP from 6.2 nM to ~1 μM affinity (*Assi et al., 2010*). While this did not alter the protein expression (*Figure 1—figure supplement 2C,D*), there was an expected BRET signal reduction (*Figure 1—figure supplement 2C*). Indeed, it significantly increased the BRET$_{50}$ (an approximation of the relative affinity of the acceptor fusion for the donor fusion proteins, corresponding to the acceptor/donor ratio necessary to reach 50% of the BRET$_{max}$) and significantly reduced the BRET$_{max}$ (an approximation for the total

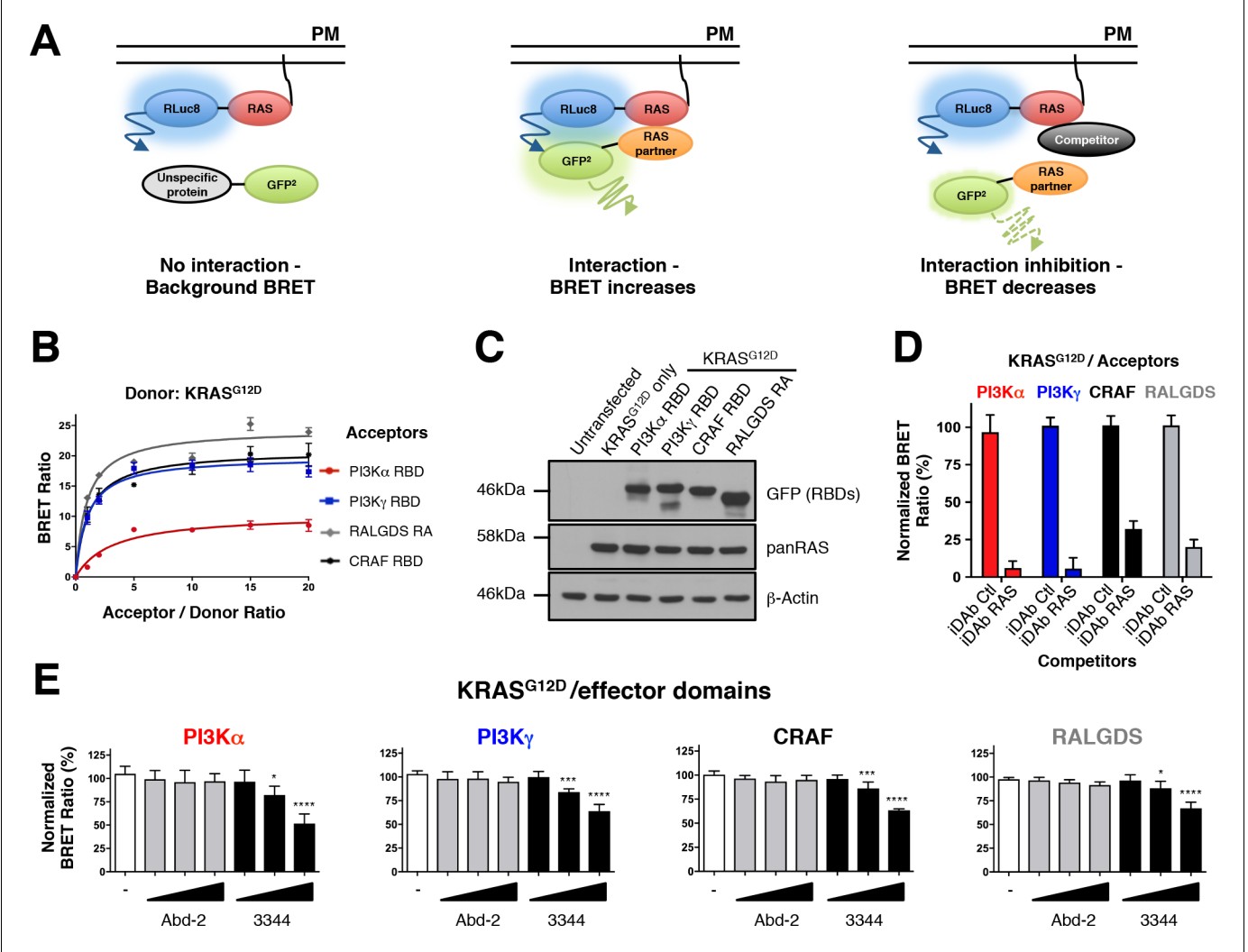

**Figure 1.** RAS-effector BRET biosensors and interference of KRAS-effector interactions by a RAS-binding compound. An outline of the BRET2-based RAS biosensor system is shown in **A**. RAS bound to the plasma membrane (PM) is fused at its amino terminal end to the RLuc8 moiety (donor). When a protein fused to the GFP² moiety (acceptor) does not bind to RAS, it only produces a background BRET signal. However, when an acceptor binds to RAS, it induces a BRET signal, if the luciferase and GFP domains are within 100 Å. The BRET signal can be decreased by addition of a competitor (either by a macrodrug or a small molecule inhibitor). The interaction titration of full-length KRAS$^{G12D}$-CAAX (for simplicity, the CAAX motif is omitted in all the RAS constructs described hereafter) with the four effector acceptor proteins and the effect on intracellular protein levels are shown in **B** and **C**. Competition assays show the specificity of the RAS biosensors in **D** (iDAb) and **E** (RAS-binding compounds). In **D**, the non-relevant anti-LMO2 iDAb (called hereafter iDAb control, Ctl) serves as a negative control and anti-RAS iDAb (herein named iDAb RAS) serves as a positive control. In **E**, 3344 (black bars) decreases KRAS$^{G12D}$/effector domain interactions in a dose-dependent manner showing its broad range of inhibition. Cells were treated with 5, 10 and 20 µM of 3344 (black bars), Abd-2 (grey bars) or DMSO alone (white bars) as the negative control. Statistical analysis was performed with a one-way ANOVA followed by Dunnett's post-hoc tests (*p<0.05, ***p<0.001, ****p<0.0001). Each experiment was repeated three (**B, D**) or four times (**E**). Where error bars are presented, these correspond to mean values ± SD of biological repeats (**B, D–E**). See also *Figure 1—figure supplement 1*, *Figure 1—figure supplement 2*, *Figure 1—figure supplement 3* and *supplementary file 1*.

DOI: https://doi.org/10.7554/eLife.37122.003

The following figure supplements are available for figure 1:

**Figure supplement 1.** Optimization of the RAS biosensors.

DOI: https://doi.org/10.7554/eLife.37122.004

**Figure supplement 2.** Validation of the RAS biosensors with the anti-iDAb RAS.

DOI: https://doi.org/10.7554/eLife.37122.005

**Figure supplement 3.** 3344 inhibits RAS-RBD interactions.

DOI: https://doi.org/10.7554/eLife.37122.006

number of complex RAS/iDAb and the distance between the donor and the acceptor within the dimer), which together are consistent with a decreased affinity of this mutant iDAb toward RAS. Therefore, the results obtained with the iDAb RAS confirmed the sensitivity and accuracy of the RAS biosensors.

Finally, we tested the inhibition of interaction between RAS and its effector partners using BRET in a competition assay. HEK293T cells were transiently transfected with KRAS$^{G12D}$, each of the RAS-effector domain and a competitor (non-GFP$^2$) version of the iDAb RAS or iDAb control. This competition showed that iDAb RAS, but not the control, drastically decreased the BRET ratio of all the interactions tested (*Figure 1D*). These results confirmed that the BRET2 biosensors enable monitoring of PPI inhibition of KRAS$^{G12D}$ with each of the four effectors tested by the anti-RAS single domain antibody.

## The BRET2 biosensors show that 3344 is an inhibitor of KRAS-effector interactions

Our major purpose in the development of the RAS BRET2 biosensors was to create a validation tool for compounds that bind to RAS and interfere with its PPI in living cells. We have identified compounds that bind to KRAS using in vitro screening and one compound 3344 (chemical structure and 1-D NMR characterization shown in *Figure 1—figure supplement 3A–C*) binds to KRAS$^{G12V}$ with an affinity of 126 nM using $^1$H Carr-Purcell-Meiboom-Gill (CPMG) NMR (*Baldwin and Kay, 2009*) (data are shown in *Figure 1—figure supplement 3D*). In vitro competition studies of 3344 binding to KRAS$^{G12V}$ in waterLOGSY NMR show the anti-RAS scFv inhibits 3344 binding to KRAS (*Figure 1—figure supplement 3E*). In view of the in vitro inhibition by the anti-RAS scFv of 3344 binding to RAS and because the iDAb RAS interferes with BRET signal in cells (*Figure 1D*), 3344 was used for validation of the BRET2 toolbox for RAS-effector PPI inhibitors. In the subsequent experiments reported here, we compare 3344 with an initial compound (Abd-2) obtained through a SPR in vitro screening, which binds HRAS/KRAS with low affinity. It is the precursor of the 3344 compound and both share the same benzodioxane group (the structures of 3344 and Abd-2 are shown in *Figure 1—figure supplement 3A,F*). These compounds have been selected from a medicinal chemistry programme in order to validate the BRET-based RAS biosensors.

HEK293T cells were transiently transfected with BRET pairs and, after 24 hr to allow protein expression, the cells were seeded in 96-well plates. The compounds were added at different concentrations (5, 10 and 20 μM) and incubated on cells for a further 20 hr before the BRET reading. For each assay, the donor protein was RLuc8-KRAS$^{G12D}$ and the acceptor proteins were PI3Kα RBD-GFP$^2$, PI3Kγ RBD-GFP$^2$, CRAF RBD-GFP$^2$ or RALGDS RA-GFP$^2$. We observed a dose response reduction in BRET signal for the assays with compound 3344 but not with the Abd-2 indicating that only 3344 interferes with the RAS-effector PPI (*Figure 1E*). To rule out the possibility of false positive compounds (for instance, that might interfere directly with the BRET signal), we included control BRET-based biosensors. We tested the RAS compounds with the iDAbs RAS biosensors, either with RLuc8-LMO2 donor and iDAb$_{dm}$ LMO2 (a dematured anti-LMO2 iDAb (*Sewell et al., 2014*)) acceptor (*Figure 1—figure supplement 3G*), RLuc8-KRAS$^{G12D}$ donor with the iDAb RAS acceptor (*Figure 1—figure supplement 3H*), or RLuc8-KRAS$^{G12D}$ donor with the iDAb$_{dm}$ RAS acceptor (*Figure 1—figure supplement 3I*). Abd-2 has no effect on any of these assays while 3344 only interferes, in a dose response, with KRAS$^{G12D}$/iDAb$_{dm}$ RAS-induced BRET without affecting the expression of the biosensors (*Figure 1—figure supplement 3J*). Hence, the inhibitory effects of 3344 on KRAS$^{G12D}$-effectors interactions are not simply due to interference with the BRET assay.

## BRET2 reporter and associated RAS-CRAF signaling are affected by compound 3344

The RAS binding domain of the effector molecules lack some regulatory domains, which impedes a direct study of RAS inhibitors on pathways downstream of RAS. To reduce this limitation, we developed an optimized RAS biosensor of the full-length CRAF$^{S257L}$ mutant (herein named CRAF$^{FL}$) since the S257L mutation increases ERK phosphorylation (*Razzaque et al., 2007*) and because we found that CRAF$^{FL}$ interacts with KRAS$^{G12D}$ but not with KRAS$^{S17N}$ (*Figure 2—figure supplement 1A*). We performed a competition assay with the iDAb RAS confirming that it impedes the BRET2 signal due to the binding of CRAF$^{FL}$ with KRAS$^{G12D}$, in a dose response mode, whereas the iDAb control had

no effect (*Figure 2A*). There was no alteration in CRAF$^{FL}$ and KRAS$^{G12D}$ protein expression due to the transfection of the iDAbs, shown by western analysis (*Figure 2—figure supplement 1B*). In addition, iDAb RAS inhibition significantly decreased the phosphorylation of MEK1/2 and ERK1/2 kinases (*Figure 2B* shows western blot data, quantitated in *Figure 2C*), confirming results affecting endogenous ERK phosphorylation by iDAb RAS interaction with RAS (*Tanaka and Rabbitts, 2010*).

We further tested the ability of the small molecule 3344 to inhibit the KRAS$^{G12D}$/CRAF$^{FL}$ biosensor and the downstream biomarker pathways with either a long incubation (20 hr, *Figure 2D–F*) or a short incubation (3 hr, *Figure 2—figure supplement 1D–F*) to further validate the specificity of

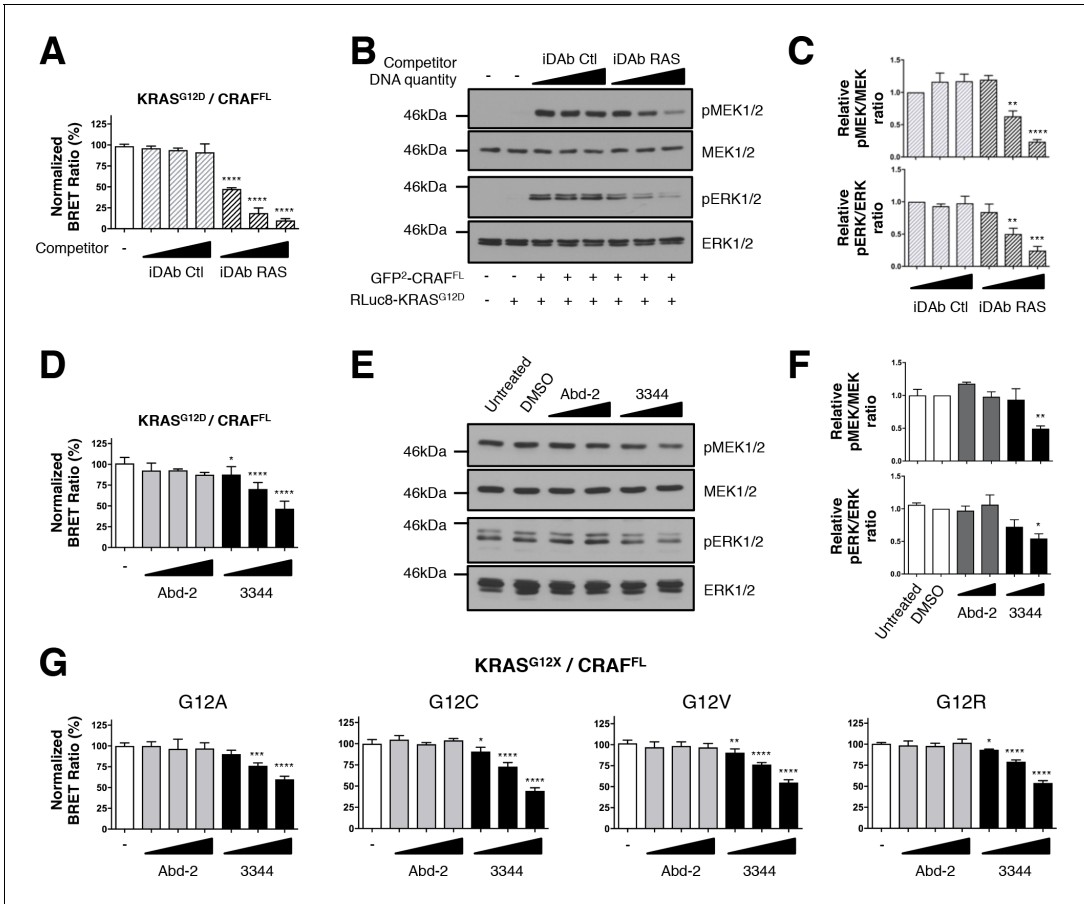

**Figure 2.** BRET biosensors of KRAS$^{G12}$ mutants and full-length CRAF are inhibited by compound 3344. A biosensor for the full-length CRAF$^{S257L}$ (CRAF$^{FL}$) protein was made and tested for interaction with mutants of KRAS glycine 12. For **A** and **B**, the plasmids expressing BRET pair KRAS$^{G12D}$/ CRAF$^{FL}$ was transfected into HEK293T cells and competed with iDAb expression as indicated; the BRET ratios are shown in **A** and western blot data in **B**. The iDAb RAS inhibition of phosphorylation of ERK and MEK signals are quantified in **C**. The β-actin loading control, iDAbs and BRET pair expression controls are shown in *Figure 2—figure supplement 1*. In **D**, the BRET ratio of KRAS$^{G12D}$/CRAF$^{FL}$ interaction was measured in the presence of an increasing dose of compound 3344. This induces a dose-dependent decrease of MEK and ERK kinase phosphorylation (**E**) after cells expressing the KRAS$^{G12D}$/CRAF$^{FL}$ biosensor pair were treated 20 hr with DMSO, 10 and 20 µM of Abd-2 and 3344 compounds or not treated (untreated lane). The β-actin loading control and BRET pair expression controls are shown in *Figure 2—figure supplement 1*. Quantification of the relative levels of pMEK1/ 2 and pERK1/2, normalized to total MEK1/2 and ERK1/2 respectively, are shown in **F**. The RAS biosensor toolkit includes KRAS G12A, G12C, G12V and G12R, in addition to KRAS G12D. In **G**, each was expressed with CRAF$^{FL}$ and BRET ratios determined at 0, 5, 10 and 20 µM Abd-2 or 3344. Statistical analyses in **C** were performed using a one-way ANOVA followed by Sidak's post-hoc tests and in **A**, **D**, **F** and **G** using a one-way ANOVA followed by Dunnett's post-tests (*$p<0.05$, **$p<0.01$, ***$p<0.001$, ****$p<0.0001$). Each experiment was repeated twice (**E–F**), three times (**B–D**), four times (**A**) or five times (**G**). Where error bars are presented, they correspond to mean values ± SD of biological repeats (**A**, **D**, **G**) or correspond to mean ±SEM of biological repeats (**C**, **F**). See also *Figure 2—figure supplement 1*.

DOI: https://doi.org/10.7554/eLife.37122.007

The following figure supplement is available for figure 2:

**Figure supplement 1.** Interactions of KRAS$^{G12X}$ mutants and full-length CRAF are inhibited by 3344.

DOI: https://doi.org/10.7554/eLife.37122.008

inhibition. Indeed, long-term incubation with the compound may indirectly inhibit RAS downstream pathways by affecting autocrine mechanisms involved in secondary activation of RAS pathways (*Arthur and Ley, 2013*; *Zhang et al., 2011*). We compared the effect of Abd-2 and 3344 on the BRET pair and found a significant decrease in BRET signal with 3344 that occurred in a dose-dependent manner (*Figure 2D* and *Figure 2—figure supplement 1D*) without modifying RAS or CRAF expression (as shown by western analysis, *Figure 2—figure supplement 1C,G*). Western blots using anti-pMEK and anti-pERK showed that 3344 also significantly inhibited MEK1/2 and ERK1/2 phosphorylation whilst Abd-2 did not (*Figure 2E*, quantified in *Figure 2F* and *Figure 2—figure supplement 1E–F*). Therefore, these observations show a specific and functional effect of the inhibition of interaction between RAS and CRAF$^{FL}$ by the 3344 with a long and short incubation.

Some compounds have been previously characterized that bind selectively on the cysteine of KRAS$^{G12C}$ mutant (*Lito et al., 2016*; *Ostrem et al., 2013*; *Patricelli et al., 2016*). We assessed whether our compound 3344 was able to interfere with binding of a range of mutant KRAS Gly12 proteins, including G12C, with CRAF in BRET assays. Analysis of the BRET2 signals from interaction of KRAS$^{G12A}$, KRAS$^{G12C}$, KRAS$^{G12V}$ and KRAS$^{G12R}$ with CRAF$^{FL}$ showed a dose response effect of compound 3344 but not Abd-2 (*Figure 2G*). The corresponding BRET biosensor acceptor and donor proteins are equally expressed after transfection as judged by western blot analysis (*Figure 2—figure supplement 1H*).

Therefore, using this new set of validated RAS biosensors, we show that the compound disrupts mutant KRAS/CRAF$^{FL}$ interaction in cells. In turn, this leads to inhibition of the RAF/MEK/ERK downstream signaling pathway (that emanates from the transfected protein expression).

## 3344 inhibits the wild type KRAS-CRAF biosensor and its downstream signaling pathway

We extended the repertoire of biosensors by analyzing wild-type KRAS (KRAS$^{WT}$) donor molecule and also assessed if epidermal growth factor (EGF)-stimulated MEK/ERK phosphorylation (*Burgering et al., 1993*; *Lange-Carter and Johnson, 1994*) could be altered through the interaction of a KRAS$^{WT}$/CRAF$^{FL}$ BRET2 biosensor protein pair. Although the iDAb RAS binds weakly to RAS$^{WT}$ in transfected mammalian two-hybrid reporter cells (*Tanaka et al., 2007*), we first established if the BRET2 signal from RLuc8-KRAS$^{WT}$ and GFP$^2$-CRAF$^{FL}$ PPI could be inhibited by the iDAb RAS in the BRET transfection assay. HEK293T cells were transfected with the BRET pair and serum was removed for 24 hr, stimulated for 5 min with EGF and the BRET ratio directly determined after the stimulation. EGF treatment brings KRAS$^{WT}$ and CRAF$^{FL}$ fusion proteins in a closer proximity and enhances the number of KRAS$^{WT}$/CRAF$^{FL}$ dimers because the BRET$_{max}$ value increases from 4.02 to 10.01 (*Figure 3—figure supplement 1A*). A dose response inhibition of the BRET2 signal was observed with iDAb RAS, but not iDAb control (*Figure 3A*), which correlated with the reduction of pMEK1/2 and pERK1/2 detected by western blots (*Figure 3B* and quantified in *Figure 3C*). This shows that the RAS BRET2 biosensors can be used to couple PPI effects and signaling effects.

We conducted parallel BRET2 dose response experiments with the 3344, compound compared to Abd-2, implementing EGF stimulation and using the KRAS$^{WT}$/CRAF$^{FL}$ biosensor with short and long incubation times (3 hr and 20 hr, respectively). Compound 3344 inhibits this interaction in a dose-response manner (*Figure 3D* and *Figure 3—figure supplement 1D*) and prevents the phosphorylation of MEK1/2 and ERK1/2 kinases (*Figure 3E*, quantified in *Figure 3F* and *Figure 3—figure supplement 1E–F*). Protein levels per se were not affected by the BRET2 transfectants by either the iDAb expression (*Figure 3—figure supplement 1B*) or Abd-2 or 3344 treatments (*Figure 3—figure supplement 1C,G*). In conclusion, use of the 3344 with the BRET2 RAS biosensors confirms this compound is a pan-KRAS-effector PPI inhibitor.

## 3344 inhibits the RAS-PI3K-AKT signaling pathway

We have also explored the second best-characterized RAS effector family, the RAS-PI3Kα-AKT pathway (*Castellano and Downward, 2011*) by establishing a KRAS$^{G12D}$/full-length PI3Kα (herein PI3Kα$^{FL}$) biosensor. In this case, we required a tripartite system as we observed that co-expression of the p85α regulatory subunit with PI3Kα$^{FL}$-GFP$^2$ was required to obtain detectable, specific and optimized BRET signal from interaction of KRAS$^{G12D}$ and PI3Kα$^{FL}$ (*Figure 4—figure supplement 1A*). KRAS$^{S17N}$ mutant showed no specific interaction with PI3Kα$^{FL}$ further confirming the accuracy

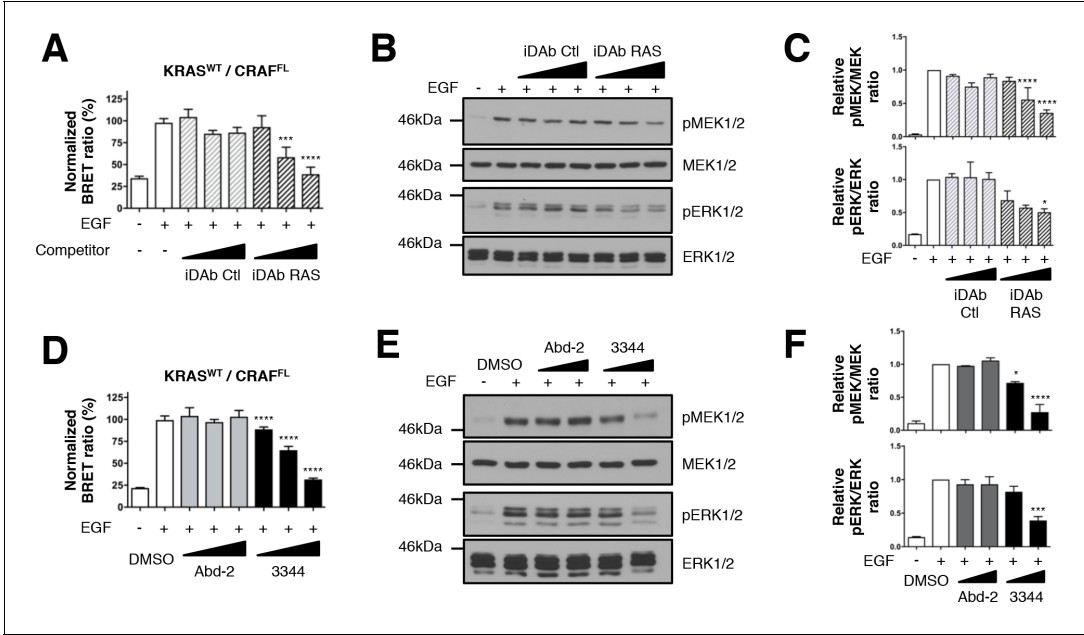

**Figure 3.** Wild-type KRAS and CRAF biosensor interaction-induced signaling is impaired by 3344. The BRET KRAS$^{WT}$/CRAF$^{FL}$ pair was tested for interaction after EGF stimulation of HEK293T cells in presence of competitors. In **A**, cells were transfected with plasmids to express the KRAS$^{WT}$ biosensor with or without iDAbs and stimulated by EGF (50 ng/mL). iDAb RAS shows an inhibition of KRAS$^{WT}$/CRAF$^{FL}$ interaction after EGF treatment in a dose-dependent manner. **B** is a western blot of the transfected cells from panel A showing the effect of the iDAbs on EGF-stimulated RAS-RAF-MEK-ERK signaling pathway (pMEK and pERK signals are quantified in **C**). β-actin loading control, iDAbs and BRET pair expression controls are shown in *Figure 3—figure supplement 1*. The effect on BRET2 signal of compounds Abd-2 (grey bars) and 3344 (black bars) on KRAS$^{WT}$/CRAF$^{FL}$ interaction after EGF treatment in a BRET competition experiment is shown in panel D. In panel E, HEK293T cells were transfected as in **D** with the plasmids expressing the BRET pair KRAS$^{WT}$/CRAF$^{FL}$ for 24 hr and serum starved 20 hr in the presence of DMSO, 10 and 20 μM of Abd-2 and 3344 compounds. Cells were treated 5 min with EGF (50 ng/mL), lysed and analyzed by western blot. The expression level of the BRET protein pair is shown in *Figure 3—figure supplement 1* as well as the loading control β-actin for the western blot. The western blot data are quantified in panel F. One-way ANOVA followed by Dunnett's post-hoc tests were used to determine the statistical significance of BRET, pERK and pMEK modulations induced by the compound or the iDAb (*p<0.05, ***p<0.001, ****p<0.0001). Each experiment was repeated twice (**B–C**) or three times (**A, D–F**). Where error bars are presented, they correspond to mean values ± SD of biological repeats (**A, D**) or correspond to mean ±SEM of biological repeats (**C, F**). See also *Figure 3—figure supplement 1*.

DOI: https://doi.org/10.7554/eLife.37122.009

The following figure supplement is available for figure 3:

**Figure supplement 1.** 3344 inhibits KRAS$^{WT}$/CRAF$^{FL}$ interaction induced by EGF treatment.

DOI: https://doi.org/10.7554/eLife.37122.010

of this biosensor (*Figure 4—figure supplement 1A*). We validated the BRET biosensor by showing that the iDAb RAS impaired that interaction in a dose-dependent manner, whereas the iDAb control did not (*Figure 4A*). Western blot analysis showed some reduction in PI3K and RAS proteins, specifically concordant with expression of the iDAb RAS (*Figure 4—figure supplement 1B*) and there was also a dose response reduction of phosphorylation of the downstream biomarker AKT at Ser473 (*Figure 4B* and quantified in *Figure 4C*).

Implementing the same biosensor assay treated with the compound 3344 for 3 or 20 hr, we confirmed this compound interferes with the KRAS$^{G12D}$/PI3Kα$^{FL}$ interaction (*Figure 4D–F* and *Figure 4—figure supplement 1D–F*) without loss of protein (*Figure 4—figure supplement 1C,G*). Abd-2 has no effect on the phosphorylation of AKT that results from KRAS$^{G12D}$/PI3Kα$^{FL}$ interaction. Conversely, 3344 does affect RAS-PI3K interaction and AKT phosphorylation. When increasing doses of either Abd-2 or 3344 were used in the BRET-transfected cells, we observed dose response reduction of BRET signal with 3344 but not Abd-2 (*Figure 4D* and *Figure 4—figure supplement 1D*). Associated with this inhibition, was a reduction in the downstream biomarker AKT Ser473 phosphorylation (*Figure 4E*, quantified in *Figure 4F* and *Figure 4—figure supplement 1E–F*). 3344 inhibits RAS-PI3Kα PPI and thus signaling through AKT.

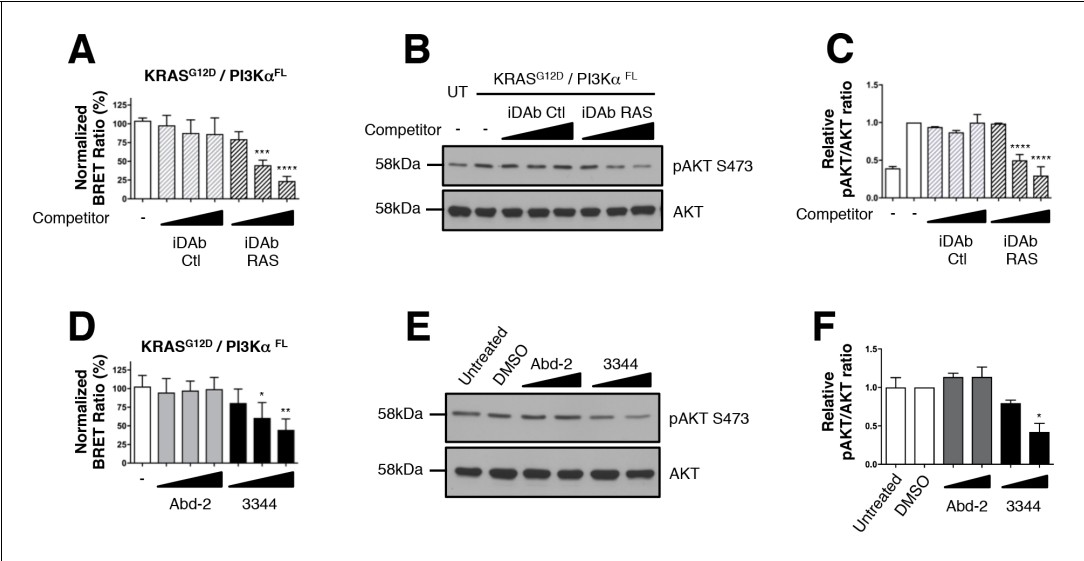

**Figure 4.** Interaction between mutant KRAS and full-length PI3Kα BRET pair interaction is impeded by 3344. The BRET signal produced from the interaction of the KRAS$^{G12D}$ and full-length PI3Kα (PI3Kα$^{FL}$) was obtained by transfecting HEK239T cells with plasmids encoding this BRET pair. In **A**, cells were co-transfected with the biosensor and increasing levels of competitor plasmids encoding iDAbs RAS (black striped bars) or iDAb control (grey striped bars) or biosensor alone (white bar). iDAb RAS impedes KRAS$^{G12D}$/PI3Kα$^{FL}$ interaction and this inhibition causes a decrease of pAKT at serine 473 as shown by western blot in **B** and its quantification in **C**. UT is for untransfected cells. In **D**, HEK293T cells transfected with the BRET biosensor KRAS$^{G12D}$/PI3Kα$^{FL}$ were treated for 20 hr with DMSO (white bar), 5, 10 and 20 µM of Abd-2 (grey bars) and 3344 (black bars) compounds and the BRET signal of the biosensor was assessed. In panel **E**, the cells were transfected and treated as in **D** but with 10 and 20 µM of Abd-2 and 3344 compounds. 20 hr after the treatment, cells were lysed and analysed by western blot using anti-pAKT (Ser 473) or anti-pan-AKT antibody. The signal in the western blot is quantitated in **F**. Related controls are shown on *Figure 4—figure supplement 1*. One-way ANOVA followed by Dunnett's post-hoc tests were used to determine the statistical significance of BRET and pAKT modulations induced by the compound or the iDAb (*$p < 0.05$, **$p < 0.01$, ***$p < 0.001$, ****$p < 0.0001$). Each experiment was repeated twice (**E–F**) or three times (**A–D**). Where error bars are presented, they correspond to mean values ± SD of biological repeats (**A, D**) or correspond to mean ±SEM of biological repeats (**C, F**). See also *Figure 4—figure supplement 1*.
DOI: https://doi.org/10.7554/eLife.37122.011

The following figure supplement is available for figure 4:

**Figure supplement 1.** Interaction of KRAS$^{G12D}$ with PI3Kα$^{FL}$ is inhibited by 3344.
DOI: https://doi.org/10.7554/eLife.37122.012

## The BRET2 biosensor toolbox includes NRAS and HRAS and shows 3344 inhibits PPI of the RAS family

The KRAS, NRAS and HRAS family members are conserved proteins that have an almost identical amino-acid domain (G domain) from residues 1–166 but a C-terminal hypervariable domain (*Wennerberg et al., 2005*). We have extended the RAS biosensor toolbox to include NRAS and HRAS. We used full-length NRAS$^{Q61H}$ and HRAS$^{G12V}$ mutants to build these new RAS biosensors for use with the various effector RBDs. These mutants were used at the positions Q61 and G12, for NRAS and HRAS respectively, as these are the positions most frequently mutated in human cancer involving NRAS and HRAS mutants (*Cox et al., 2014*). Titration of the RAS donor and CRAF$^{FL}$ acceptor proteins show that the RLuc8-NRAS$^{Q61H}$ and RLuc8-HRAS$^{G12V}$ proteins interact and reach plateau BRET signals with GFP$^2$-CRAF$^{FL}$ (*Figure 5—figure supplement 1A*). Furthermore, the BRET2 signal is diminished by increasing levels of the iDAb RAS but not the iDAb control (*Figure 5—figure supplement 1B–D*) as expected from the analysis of the effects of the anti-RAS intracellular antibody (*Tanaka and Rabbitts, 2010*; *Tanaka et al., 2007*).

We further evaluated the efficacy of the RAS-binding compounds Abd-2 and 3344 in binding to NRAS and HRAS using a BRET assay in which the RAS protein donors were co-expressed with either PI3K, CRAF or RALGDS acceptors (*Figure 5A–D*). While the low-affinity Abd-2 compound does not interfere with the BRET signal in any of the NRAS and HRAS BRET assays using either effector RBDs (*Figure 5A,B*) or full-length CRAF (*Figure 5C,D*), the compound 3344 disturbs the BRET2 signal in

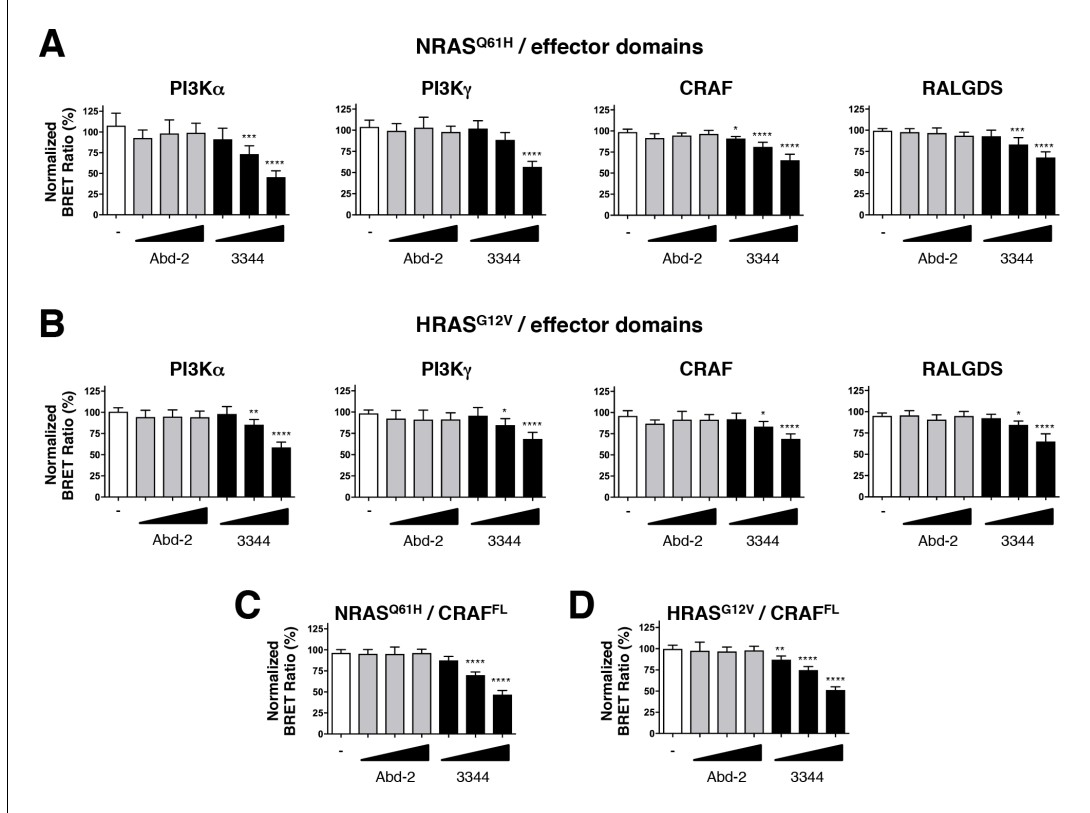

**Figure 5.** Compound 3344 inhibits NRAS and HRAS-effector BRET-based biosensors. HEK293T cells were transfected 24 hr with plasmids expressing the NRAS[Q61H] (A, C) and HRAS[G12V] (B, D) biosensors together with the indicated RBDs of PI3K, CRAF and RALGDS (A, B) or full-length CRAF (C, D). These were treated with 5, 10 and 20 μM of Abd-2 (grey bars) or 3344 (black bars) compounds for 20 hr. DMSO (white bar) was used as the negative control. Statistical analyses were performed using a one-way ANOVA followed by Dunnett's post-tests (*p<0.05, **p<0.01, ***p<0.001, ****p<0.0001). Each experiment was repeated at least four times. Where error bars are presented, they correspond to mean values ± SD of biological repeats (A–D). See also *Figure 5—figure supplement 1*.

DOI: https://doi.org/10.7554/eLife.37122.013

The following figure supplement is available for figure 5:

**Figure supplement 1.** iDAb RAS inhibits mutant NRAS and HRAS interaction with CRAF[FL].

DOI: https://doi.org/10.7554/eLife.37122.014

a dose-response manner in all these RAS interactions (*Figure 5* and *Figure 5—figure supplement 1E,F*). Therefore, the BRET-based RAS biosensors characterization of 3344 shows this compound as a pan-RAS-effector interactions inhibitor that binds KRAS, NRAS and HRAS.

## Compound 3344 binds to a pocket close to the switch regions of mutant KRAS

The implementation of our RAS BRET2 toolbox showed that the compound 3344 is able to bind the transfected RAS protein products at the plasma membrane and interfere with their effector interaction. In addition, the downstream signaling was impeded. The mechanism of the interaction inhibition was corroborated by X-ray crystallography of KRAS[Q61H] soaked with compound 3344. *Figure 6A* shows that 3344 binds to KRAS in a previously identified pocket (*Maurer et al., 2012*; *Sun et al., 2012*) close to the switch regions where the effectors interact with RAS (*Table 1* has the refinement statistics for the X-ray data). The superimposition of the structures of three RAS-effector protein complexes with the structure of KRAS-3344 complex shows that parts of 3344 would overlap with the bound effector structures, suggesting that the competition effect of 3344 can be explained by straightforward steric hindrance (*Figure 6B*). We further confirmed that 3344 could interfere with the endogenous RAS-effector PPI in two human cancer cell lines (viz. colorectal adenocarcinoma DLD-1 cells expressing KRAS[G13D] and non-small cell lung carcinoma H358 cells expressing

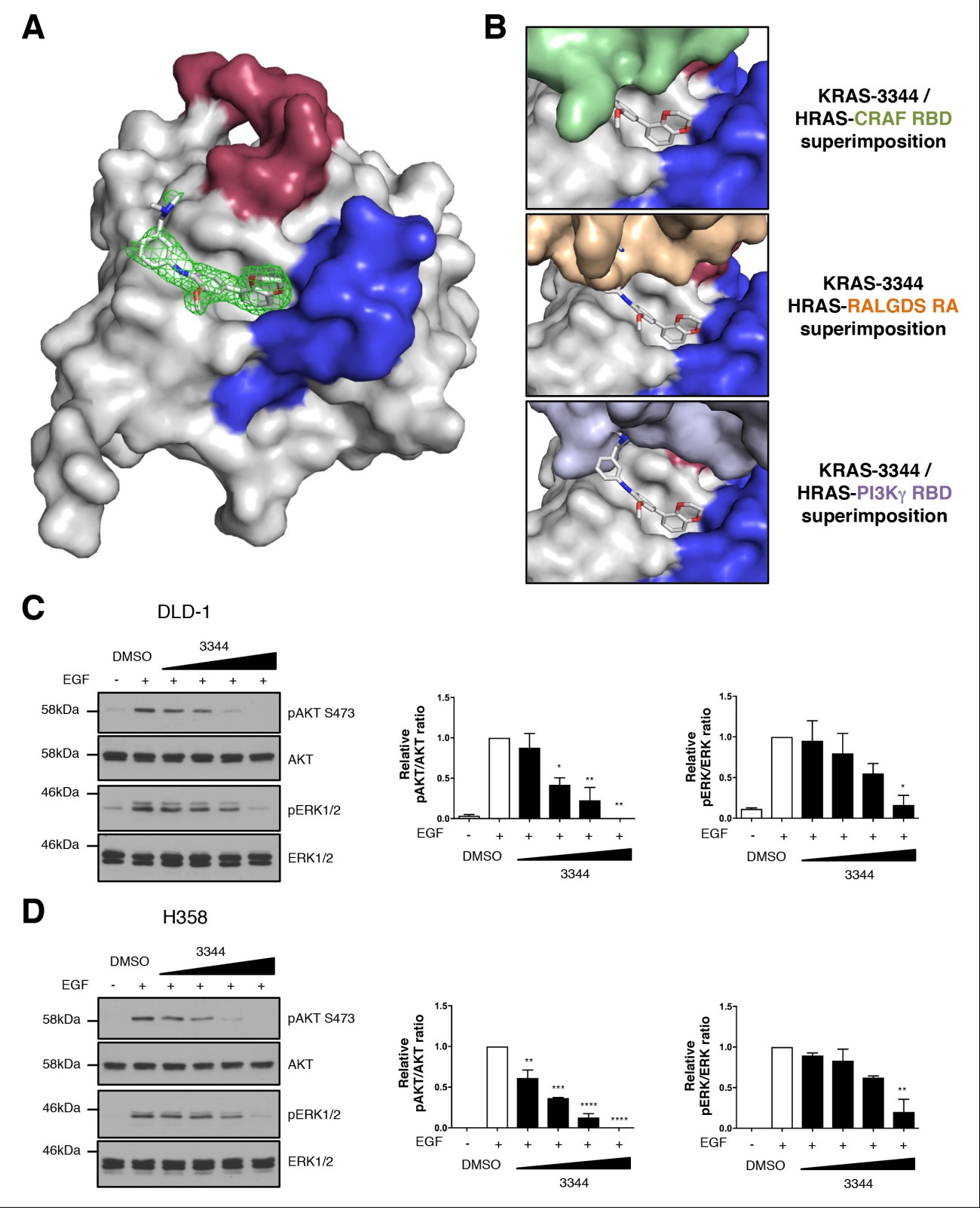

**Figure 6.** Compound 3344 interacts in a pocket close to the switch regions of KRAS. The interaction of mutant KRAS with compound 3344 was analyzed by X-ray crystallography. (**A**) KRAS$^{Q61H}$ crystals were soaked with 3344 compound and crystal structures obtained from X-ray diffraction. The compound is shown binding in the hydrophobic pocket near switch I (shown in red) and switch II (shown in blue). The electron density map of the compound (2Fo-Fc) is shown as green mesh, and contoured at 1.0 rms. (**B**) We have modeled the potential interactions that could prevent 3344 and a

*Figure 6 continued on next page*

*Figure 6 continued*

RAS effector binding simultaneously to the same RAS molecule by overlaying our structure of the KRAS-3344 complex onto the published structures of top panel: HRAS-CRAF RBD (PDB 4G3X), middle panel: HRAS-RALGDS RA (PDB 1LFD), bottom panel: HRAS-PI3Kγ RBD (PDB 1HE8). (**C, D**) Two human mutant KRAS expressing lines (C: DLD-1 and D: H358) were serum-starved for 24 hr and treated 3 hr with different concentrations of 3344 (2, 5, 10 and 20 μM) before stimulation with EGF (50 ng/mL) for 10 min. Cells were harvested, proteins extracted and separated by SDS-PAGE for western blot analysis. Western membranes were treated with anti-pAKT S473; anti-pan AKT; anti-pERK1/2 and anti-ERK1/2 as indicated. Statistical analyses of pERK/ERK and pAKT/AKT quantifications were performed using a one-way ANOVA followed by Dunnett's post-tests (*$p < 0.05$, **$p < 0.01$, ***$p < 0.001$, ****$p < 0.0001$). Where error bars are presented, they correspond to mean values ± SEM of biological repeats (**C–D**). Each experiment was performed twice (**C–D**).

DOI: https://doi.org/10.7554/eLife.37122.015

KRAS[G12C]). The cells were serum starved 24 hr and stimulated 10 min with EGF in the presence of increasing amounts of 3344, followed by western blot protein analysis to detect phosphorylated AKT Ser473 or phosphorylated ERK (*Figure 6C,D*). 3344 decreases EGF-induced pAKT and pERK1/2 abundance in both cell types with an observed $IC_{50}$ of ~5–10 μM without any change in the total levels of AKT or ERK1/2. Therefore, 3344 can interfere with endogenous RAS signaling in human cancer cell lines. As our BRET2 results show direct interference of RAS-effector PPI by 3344, we conclude that this is the mechanism of inhibition of the biomarkers in the tumor cell assay.

## Discussion

BRET-based biosensors have been successfully used to discover and characterize small molecules inhibitors (*Beautrait et al., 2017*; *Corbel et al., 2011*; *Lavoie et al., 2013*; *Mazars and Fåhraeus, 2010*; *Robinson et al., 2014*). The development of such biosensors involves the optimization of multiple parameters such as the fusion position of the RLuc8 and GFP[2] moieties on their respective protein N- or C-terminus and the determination of the appropriate quantity of donor and acceptor plasmids for intracellular expression. Notably, the latest parameter has to be optimized in order to avoid the titration of active compounds if transient protein expression is used (*Couturier and Deprez, 2012*). In this study, we have engineered and optimized a complete set of RAS biosensors that includes several different mutant forms of KRAS and other family members (viz. mutant NRAS and HRAS). This toolbox allows the monitoring of RAS-effector interactions and the assessment of RAS PPI inhibition by a macrodrug (iDAb RAS) and 3344, a new anti-RAS small molecule derived from an intracellular antibody fragment, in living cells. Furthermore, when the full-length biosensors were used, we could couple the RAS PPI inhibition to the signaling effects, thereby providing additional insights into the behavior of RAS inhibitors.

The inhibition of RAS PPI by 3344 in cells was demonstrated by the RAS biosensors toolbox and validated by X-ray crystallography. 3344 binds to a hydrophobic pocket near to the effector-binding switch regions of RAS (*Figure 6*). Whereas 3344 does not make direct contact with the switch regions, the BRET data show that the binding geometry and potency of 3344 is sufficient to interfere with the interaction of RAS-effector molecules that bind close to the 3344 site.

While the RAS biosensors rely on transfection and expression of RAS with one of its partner proteins rather than observations of endogenous proteins, it nevertheless offers several advantages for the study of RAS-effector interactions inhibition. It provides a direct and quantitative measurement of the PPI interference with inhibitors (i.e. small molecules or macrodrugs), which could allow the comparison of different compounds (e.g. for structure-activity-relationship studies) or macrodrugs and therefore the selection of more potent inhibitors. It is also sensitive and consequently requires a small quantity of cells to study the inhibition of the interaction. Nonetheless, 3344 prevents endogenous RAS-dependent signaling in two different human tumor cell lines at a lower concentration ($IC_{50}$ around 5 μM) (*Figure 6C,D*) than in the BRET assay with observed $IC_{50}$ around 20 μM. This difference probably reflects the expression levels of the target proteins in the two assays, where the BRET2 assay relies on transient transfection. Indeed, the overexpression in HEK293T cells probably produces higher amount of mutant RAS/effector proteins than the endogenous counterparts in cancer cells. Therefore, it might be more difficult to quantitatively inhibit the exogenous RAS/effector interaction than the endogenous one with 3344 compound. Generating stable BRET2 cell lines could minimize this difference.

**Table 1.** Data processing and refinement statistics.

| Structure | KRAS$^{Q61H}$-3344 |
|---|---|
| Data collection | |
| PDB ID | 6F76 |
| Diffraction source | ID30A-1, ESRF |
| Temperature (K) | 100 |
| Wavelength (Å) | 0.966 |
| Rotation range per image (°) | 0.05 |
| Exposure time per image (s) | 0.092 |
| Space group | P $2_12_12_1$ |
| Molecules/asymmetric unit | 6 |
| Unit cell dimensions | |
| a, b, c (Å) | 63.17, 118.19, 155.95 |
| α, β, γ (°) | 90, 90, 90 |
| Resolution range (Å) | 77.98–2.20 (2.16–2.20)* |
| Total no. of reflections | 295785 (13854) |
| Unique reflections | 65992 (2888) |
| Completeness (%) | 99.2 (87.3) |
| Multiplicity | 4.5 (4.8) |
| Rmeas(I)$^{†}$ | 0.193 (0.997) |
| Rmerge$^{‡}$ | 0.151 (0.780) |
| Rpim(I)$^{§}$ | 0.119 (0.612) |
| I/sigma | 5 (1.8) |
| CC$_{1/2}$ (%)$^{#}$ | 0.985 (0.513) |
| Refinement | |
| No. of reflections, working set | 62692 (2744) |
| No. of reflections, test set | 3300 (144) |
| Rwork/Rfree | 22.7/25.0 |
| **No. of atoms** | |
| Protein | 8400 |
| Water | 57 |
| **Average B factors (Å$^2$)** | |
| Protein | 46.8 |
| Ligand GTP | 31.9 |
| Water | 30.1 |
| RMSD | |
| Bond lengths (Å) | 0.014 |
| Bond angles (°) | 1.67 |
| Ramachandran plot | |
| Favoured regions (%) | 97.1 |
| Additionally allowed (%) | 2.9 |
| Outliers | 0 |
| MolProbity statistics | |
| Overall score | 1.11 |
| Clash score | 1.22 |
| Rotamer outliers (%) | 1.4 |

a*Values in parentheses are for data in the highest resolution shell.

†Rmeas = $\Sigma_{hkl}\{N(hkl)/[N(hkl)-1]\}^{1/2}$ $\Sigma_i|I_i(hkl)- < I(hkl)>|/$ $\Sigma_{hkl}$ $\Sigma_i I_i(hkl)$, where $I_i(hkl)$ is the intensity of reflection $hkl$. $\Sigma_i$ is the sum over all $i$ measurements of reflection $hkl$ and N($hkl$) is the multiplicity of reflection $hkl$.

‡Rmerge = $\Sigma_{hkl}$ $\Sigma_i | I_i(hkl)-<I(hkl)>| / \Sigma_{hkl}$ $\Sigma_i$ $I_i (hkl)$, where $I_i (hkl)$ is the intensity of reflection $hkl$ and $\Sigma_i$ is the sum over all I measurements of reflection $hkl$.

§Rpim= $\Sigma_{hkl}\{1/[N(hkl)-1]\}^{1/2}$ $\Sigma_i|I_i(hkl)- < I (hkl)>|/ \Sigma_{hkl}$ $\Sigma_i I_i(hkl)$, where $I_i(hkl)$ is the intensity of reflection $hkl$, $\Sigma_i$ is the sum over all $i$ measurements of reflection $hkl$ and N($hkl$) is the multiplicity of reflection $hkl$.

#$CC_{1/2}$ is Pearson's correlation coefficient between random half data sets.

DOI: https://doi.org/10.7554/eLife.37122.016

Another advantage of this toolbox has been shown by using the iDAb RAS as an acceptor within the RAS biosensors allowing a recapitulation of the published features of this intracellular single domain antibody. Therefore, the biosensors are also important tools to study RAS protein interactions in living cells and their effect on the RAS downstream pathways before being tested in cancer cell lines. RAS biosensors use should not be limited to the discovery and characterization of RAS inhibitors. Indeed, studies suggested that isoform and residue- or codon-specific RAS mutants show differences in their ability to engage effectors and signaling properties (*Hunter et al., 2015*; *Nakhaeizadeh et al., 2016*; *Yan et al., 1998*). Accordingly, RAS biosensors could also be a methodology to decipher RAS isoform/mutant properties in cells. Our toolbox is an available resource for RAS-drug development programmes, and more generally for the RAS community, since our results demonstrate the possibility of using these RAS biosensors as a generic method to characterize cell-potent RAS-binding compounds or RAS-binding macrodrugs.

The BRET2 biosensor system could also be used for direct screens of PPI inhibitors with libraries of compounds. However, because initial compounds from a library are not expected to have high affinity for their target, relatively weak interactions between donor and acceptors should be involved in the generation of BRET2 signal. This provides a further use of intracellular domain antibodies where reduction of affinity (dematuration) from a tool initially used for target validation, can be achieved to make a screening tool. Thus, the method is an approach that is transferable to other PPI situations required for drug development programmes in cancer or any other clinical indication.

## Materials and methods

**Key resources table**

| Reagent type (species) or resource | Designation | Source or reference | Identifiers | Additional information |
|---|---|---|---|---|
| Cell line (human) | HEK293T | ATCC | Cat#CRL-3216 RRID:CVCL_0063 | |
| Cell line (human) | DLD-1 | ATCC | Cat#CCL-221 RRID:CVCL_0248 | |
| Cell line (human) | H358 | ATCC | Cat#CRL-5807 RRID:CVCL_1559 | |

*Continued on next page*

*Continued*

| Reagent type (species) or resource | Designation | Source or reference | Identifiers | Additional information |
|---|---|---|---|---|
| Transfected construct (human) | pEF-RLuc8-(GGGS)$_3$-KRAS$^{G12D}$-CAAX plasmid | This paper | N/A | DNA/protein sequences provided in the *Supplementary file 1* |
| Transfected construct (human) | pEF-RLuc8-(GGGS)$_3$-KRAS$^{G12A}$-CAAX plasmid | This paper | N/A | |
| Transfected construct (human) | pEF-RLuc8-(GGGS)$_3$-KRAS$^{G12C}$-CAAX plasmid | This paper | N/A | |
| Transfected construct (human) | pEF-RLuc8-(GGGS)$_3$-KRAS$^{G12V}$-CAAX plasmid | This paper | N/A | |
| Transfected construct (human) | pEF-RLuc8-(GGGS)$_3$-KRAS$^{G12R}$-CAAX plasmid | This paper | N/A | |
| Transfected construct (human) | pEF-RLuc8-(GGGS)$_3$-NRAS$^{Q61H}$-CAAX plasmid | This paper | N/A | |
| Transfected construct (human) | pEF-RLuc8-(GGGS)$_3$-HRAS$^{G12V}$-CAAX plasmid | This paper | N/A | |
| Transfected construct (human) | pEF-RLuc8-(GGGS)$_3$-KRAS$^{S17N}$-CAAX plasmid | This paper | N/A | |
| Transfected construct (human) | pEF-RLuc8-(GGGS)$_3$-KRAS$^{WT}$-CAAX plasmid | This paper | N/A | |
| Transfected construct (human) | pEF-GFP$^2$-(GGGS)$_3$-CRAF$^{S257LFL}$ plasmid | This paper | N/A | |
| Transfected construct (human) | pEF-PI3Kα$^{FL}$-(GGGS)$_3$-GFP$^2$ plasmid | This paper | N/A | |
| Transfected construct (human) | pEF-CRAF RBD (aa 1–149)-(GGGS)$_3$-GFP$^2$ plasmid | This paper | N/A | |
| Transfected construct (human) | pEF-PI3Kα RBD (aa 161–315)-(GGGS)$_3$-GFP$^2$ plasmid | This paper | N/A | |
| Transfected construct (human) | pEF-PI3Kγ RBD (aa 190–315)-(GGGS)$_3$-GFP$^2$ plasmid | This paper | N/A | DNA/protein sequences provided in the *Supplementary file 1* |
| Transfected construct (human) | pEF-iDAb RAS-(GGGS)$_2$-GFP$^2$ plasmid | This paper | N/A | |
| Transfected construct (human) | pEF-iDAb$_{dm}$ RAS-(GGGS)$_2$-GFP$^2$ plasmid | This paper | N/A | |
| Transfected construct (human) | pEF-iDAb control-(GGGS)$_2$-GFP$^2$ plasmid | This paper | N/A | |
| Transfected construct (human) | pEF-LMO2-(GGGS)$_2$-RLuc8 plasmid | This paper | N/A | |
| Transfected construct (human) | pEF-GFP$^2$-(GGGS)$_3$-iDAb$_{dm}$ LMO2 plasmid | This paper | N/A | |
| Transfected construct (human) | pEF-memb-FLAG-iDAb RAS plasmid | This paper | N/A | |
| Transfected construct (human) | pEF-memb-FLAG-iDAb control plasmid | This paper | N/A | |
| Transfected construct (human) | pEF-iDAb RAS-myc plasmid | This paper | N/A | |
| Transfected construct (human) | pEF-iDAb control-myc plasmid | This paper | N/A | |
| Transfected construct (human) | pcDNA3.1-myc-p85α$^{FL}$ plasmid | A gift from R. Williams and O. Perisic | N/A | |
| Transfected construct (mouse) | pEF-RALGDS RA (aa 788–884)-(GGGS)$_3$-GFP$^2$ plasmid | This paper | N/A | The RALGDS RA domain corresponds to the mouse sequence |
| Antibody | Phospho-ERK 1/2 Rabbit antibody | Cell Signaling Technology | Cat#9101S RRID:AB_331646 | |

*Continued on next page*

*Continued*

| Reagent type (species) or resource | Designation | Source or reference | Identifiers | Additional information |
|---|---|---|---|---|
| Antibody | Total ERK 1/2 Rabbit antibody | Cell Signaling Technology | Cat#9102S RRID:AB_330744 | |
| Antibody | Phospho-MEK 1/2 Rabbit antibody | Cell Signaling Technology | Cat#9154S RRID:AB_2138017 | |
| Antibody | Total MEK 1/2 Mouse antibody | Cell Signaling Technology | Cat#4694S RRID:AB_10695868 | |
| Antibody | Phospho-AKT S473 Rabbit antibody | Cell Signaling Technology | Cat#4058S RRID:AB_331168 | |
| Antibody | Total AKT Rabbit antibody | Cell Signaling Technology | Cat#9272S RRID:AB_329827 | |
| Antibody | Pan-RAS Mouse antibody | Millipore | Cat#OP40 RRID:AB_213400 | |
| Antibody | GFP Mouse antibody | Santa Cruz Biotechnology | Cat#sc-9996 RRID:AB_627695 | |
| Antibody | β-Actin Mouse antibody | Sigma-Aldrich | Cat#A1978 RRID:AB_476692 | |
| Antibody | CMYC HRP-linked Goat antibody | Novus Biologicals | Cat#NB600-341 RRID:AB_10000717 | |
| Antibody | Anti-Mouse IgG HRP-linked antibody | Cell Signaling Technology | Cat#7076S RRID:AB_330924 | |
| Antibody | Anti-Rabbit IgG HRP-linked antibody | Cell Signaling Technology | Cat#7074S RRID:AB_2099233 | |
| Recombinant DNA reagent | pEF-myc-cyto vector | Invitrogen | Cat#V89120 | |
| Recombinant DNA reagent | pRLuc8-N3 vector | A gift from J. Felce | *Felce et al., 2017* | |
| Recombinant DNA reagent | pGFP$^2$-N3 vector | A gift from J. Felce | *Felce et al., 2017* | |
| Recombinant DNA reagent | pBABEpuro-CRAF$^{S257L\ FL}$ plasmid | Addgene | Addgene#51125 | |
| Peptide, recombinant protein | KRAS$^{Q61H}$ | This paper | N/A | |
| Peptide, recombinant protein | KRAS$^{G12V}$ | This paper | N/A | |
| Peptide, recombinant protein | Anti-RAS scFv | This paper | N/A | |
| Peptide, recombinant protein | Recombinant Human Epidermal Growth Factor (EGF) | Life Technologies | Cat#PHG0311 | |
| Chemical compound, drug | Coelenterazine 400a | Cayman Chemical | Cat#16157 | |
| Chemical compound, drug | 2-bromo-6-methoxyphenol | This paper | N/A | |
| Chemical compound, drug | 3-bromobenzene-1,2-diol | This paper | N/A | |
| Chemical compound, drug | 5-bromo-2,3-dihydrobenzo[b][1,4]dioxine | This paper | N/A | |
| Chemical compound, drug | 5-(4-chloro-3-methoxyphenyl)—2,3-dihydrobenzo[b][1,4]dioxine | This paper | N/A | |
| Chemical compound, drug | 4-(2,3-dihydrobenzo[b][1,4]dioxin-5-yl)-N-(4-(dimethylamino)methyl)phenyl)-2-methoxyaniline | This paper | N/A | |
| Software, algorithm | Image J | National Institutes of Health | https://imagej.nih.gov/ij/download.html RRID:SCR_003070 | |
| Software, algorithm | Prism 7.0 c | GraphPad | https://www.graphpad.com/scientific-software/prism/ RRID:SCR_002798 | |
| Software, algorithm | PROCHECK | *Laskowski et al. (1993a)* | http://www.ccp4.ac.uk/html/procheck_man/index.html | |
| Software, algorithm | REFMAC | *Murshudov et al. (1997)* | http://www.ccp4.ac.uk/html/refmac5.html RRID:SCR_014225 | |

*Continued on next page*

*Continued*

| Reagent type (species) or resource | Designation | Source or reference | Identifiers | Additional information |
|---|---|---|---|---|
| Software, algorithm | MolProbity | *Chen et al. (2010)* | http://molprobity.biochem.duke.edu/ RRID:SCR_014226 | |
| Software, algorithm | Phenix | *Adams et al. (2010)* | https://www.phenix-online.org/ RRID:SCR_014224 | |
| Software, algorithm | PyMOL | Schrodinger | https://pymol.org/2/ RRID:SCR_000305 | |
| Other | Opti-MEM I Reduced Serum Medium, no phenol red | Thermo-Fisher | Cat#11058021 | |
| Other | ViewPlate, White 96-well plate, clear bottom for tissue culture | PerkinElmer | Cat#6005181 | |
| Other | BRET2 Dual Emission optical module | PerkinElmer | Cat#2100–8140 | |
| Other | Envision instrument, Multilabel Reader | PerkinElmer | Cat#2103 | |

## Cell culture

HEK293T human embryonic kidney cells, DLD-1 cells and H358 cells were grown in DMEM medium (Life Technologies) supplemented with 10% FBS (Sigma) and 1% Penicillin/Streptomycin (Life Technologies). Cells were grown at 37°C with 5% $CO_2$ and were tested using a MycoAlert Mycoplasma Detection Kit (Lonza) and found to be mycoplasma-free before use.

## Mutation detection of RAS mutations using RT-PCR

RNA was extracted from $5 \times 10^6$ DLD-1 or H358 cells using the RNeasy Plus Mini Kit (Qiagen) according to the manufacturer's instructions. cDNA was synthesized from 1.5 to 2 μg RNA using SuperScript II Reverse Transcriptase (Invitrogen). Primers were designed to amplify KRAS DNA and incorporate HindIII and BamHI restriction sites for subcloning:

5'- TAAGCAAAGCTTATGACTGAATATAAACTTGTGGTAG-3' and

3'-GAAAATTAAAAAATGCATTATAATGTAAGGATCCTAAGCA-5'

DNA was amplified using Phusion High-Fidelity DNA Polymerase (New England Biolabs) and, following digestion with HindIII and BamHI, the DNA was cloned into pBlueScript II SK (+) (Stratagene). Plasmid DNA was prepared from indivudial DH5α transformants using a QIAprep Spin Miniprep Kit (QIAGEN). *KRAS* mutations were verified by Sanger sequencing (Source Bioscience) of at least six clones from each cell line. The *KRAS* mutations in the two human cancer cell lines were confirmed as *KRAS^G13D* in DLD-1 and *KRAS^G12C* in H358.

## Cell treatment

For dose response experiments (BRET and western blot), drugs were prepared in 100% DMSO at 10 mM. Cells were treated with Abd-2 or 3344 compounds at concentration of 5, 10 or 20 μM for 3 hr (short-term incubation) or 20 hr (long-term incubation). The compounds were diluted in the BRET medium: OptiMEM no phenol red (Life Technologies) supplemented with 4% FBS and with a final concentration of 0.2% DMSO.

For serum starvation studies with the BRET assay, cells were grown 24 hr in the presence of Opti-MEM no phenol red supplemented with 1% FBS and stimulated with 50 ng/mL EGF (Life Technologies) for 5 min at 37°C. For serum-starvation studies of cancer cell lines, cells were grown 24 hr in the presence of DMEM without FBS and stimulated 10 min with 50 ng/mL EGF. The compound was incubated for 3 hr before the EGF stimulation at 2, 5, 10 and 20 μM.

## Molecular cloning

### Generation of pEF-RLuc8 and pEF-GFP$^2$ plasmids

RLuc8 and GFP$^2$ cDNA was amplified by PCR from pRLuc8-N3 and pGFP$^2$-N3 vectors respectively (*Felce et al., 2017*). RLuc8 was cloned into the pEF-myc-cyto vector (Invitrogen) between BspHI/XhoI sites to produce a pEF-RLuc8-MCS plasmid or between NotI/XbaI sites to produce a pEF-MCS-RLuc8 plasmid. GFP$^2$ was inserted into the pEF-myc-cyto vector between NcoI/XhoI sites to produce the pEF-GFP$^2$-MCS plasmid or between NotI/XbaI to produce the pEF-MCS-GFP$^2$ plasmid. A (GGGS)$_n$ linker was introduced between XhoI/NotI of all the RLuc8 and GFP$^2$ plasmids.

### Generation of KRAS mutants and BRET donor plasmids

The generation of the mutant and wild-type KRAS was PCR site-directed mutagenesis using pPGK-KRAS$^{G12D}$-CAAX-P2A-Puro as a template (a gift from Jennifer Chambers). The following full-length KRAS mutants have been produced: KRAS$^{G12A}$, KRAS$^{G12C}$, KRAS$^{G12D}$, KRAS$^{G12V}$, KRAS$^{G12R}$, KRAS$^{S17N}$ and KRAS$^{WT}$, all with carboxy terminal CAAX. All RAS cDNAs (KRAS mutants, KRAS$^{WT}$, NRAS$^{Q61H}$ and HRAS$^{G12V}$-CAAX) were cloned between NotI/XbaI of the pEF-RLuc8-MCS plasmid.

LMO2 was amplified by PCR and cloned between NcoI/XhoI sites of the pEF-MCS-RLuc8 plasmid.

### Generation of effectors/iDAb BRET plasmids

CRAF RBD (1-149), PI3Kα RBD (161-315), full-length PI3Kα (a gift from Roger Williams and Olga Perisic), PI3Kγ RBD (190-315), RALGDS RA (788-884), iDAb RAS, iDAb$_{dm}$ RAS and iDAb LMO2 (iDAb control) were amplified by PCR and cloned between NcoI/XhoI sites of the pEF-MCS-GFP$^2$ plasmid. The full-length CRAF$^{S257L}$ was cloned between NotI/XbaI sites of pEF-GFP$^2$-MCS as well as the iDAb$_{dm}$ LMO2.

All RAS and effectors are human sequences except RALGDS RA (mouse).

All the RAS BRET constructs DNA and protein sequences have been listed in the *supplementary file 1*.

### BRET2 titration curves and competition assays

The BRET experiment protocols have been adapted from previous studies (*Lavoie et al., 2013*; *Pfleger et al., 2006*). For all BRET experiments (titration curves and competition assays) 650,000 HEK293T were seeded in each well of a six well plates. After 24 hr at 37°C, cells were transfected with a total of 1.6 μg of DNA mix, containing the donor + acceptor ± competitor plasmids, using Lipofectamine 2000 transfection reagent (Thermo-Fisher). Cells were detached 24 hr later, washed with PBS and seeded in a white 96 well plate (clear bottom, PerkinElmer) in OptiMEM no phenol red medium complemented with 4% FBS. Cells were incubated for an additional 20–24 hr at 37°C before the BRET assay reading.

### BRET2 measurements

BRET2 signal was determined immediately after addition of coelenterazine 400a substrate (10 μM final) to cells (Cayman Chemicals), using an Envision instrument (2103 Multilabel Reader, PerkinElmer) with the BRET2 Dual Emission optical module (515 nm – 30 nm and 410 nm – 80 nm; PerkinElmer). Total GFP$^2$ fluorescence was detected with excitation and emission peaks set at 405 nm and 515 nm, respectively. Total RLuc8 luminescence was measured with the Luminescence 400–700 nm-wavelength filter.

The BRET signal or BRET ratio corresponds to the light emitted by the GFP$^2$ acceptor constructs (515 nm – 30 nm) upon addition of coelenterazine 400a divided by the light emitted by the RLuc8 donor constructs (410 nm – 80 nm). The background signal is subtracted from that BRET ratio using the donor-only negative control where only the RLuc8 plasmid is transfected into the cells. The normalized BRET ratio is the BRET ratio normalized to a negative control (DMSO, no competitor or iDAb control) during a competition assay. Total GFP$^2$ and RLuc8 signals were used to control the protein expression from each plasmid.

## Western blot analysis

Cells were washed once with PBS and lysed in SDS-Tris buffer (1% SDS, 10 mM Tris-HCl pH 7.4) supplemented with protease inhibitors (Sigma) and phosphatase inhibitors (Thermo-Fisher). Cell lysates were sonicated with a Branson Sonifier and the protein concentrations determined by using the Pierce BCA protein assay kit (Thermo-Fisher). Equal amounts of protein (10 µg) were resolved on 10 or 15% SDS-PAGE and subsequently transferred onto a PVDF membrane (GE). The membrane was blocked either with 10% non-fat milk (Sigma) or 10% BSA (Sigma) in TBS-0.1% Tween20 and incubated overnight with primary antibody at 4°C. After washing the membrane was incubated with HRP conjugated secondary antibody for 1 hr at room temperature (RT, 25°C). The membrane was washed with TBS-0.1% Tween and developed using Pierce ECL Western Blotting Substrate (Thermo-Fisher) and CL-XPosure films (Thermo-Fisher). Primary antibodies include anti-phospho-p44/22 MAPK (ERK1/2) (CST), anti-p44/42 MAPK (total ERK1/2) (CST), anti-phospho-MEK1/2 (CST), anti-MEK1/2 (CST), anti-phospho-AKT S473 (CST), anti-AKT (CST), anti-pan-RAS (Millipore), anti-GFP (Santa Cruz Biotechnologies), anti-β-actin (Sigma). Secondary antibodies include anti-CMYC HRP-linked (Novus Biologicals), anti-mouse IgG HRP-linked (CST) and anti-rabbit IgG HRP-linked (CST).

## WaterLOGSY NMR

The waterLOGSY NMR method (*Dalvit et al., 2001*) was used to measure RAS ligand interaction (*Huang et al., 2017*). WaterLOGSY experiments were conducted at a $^1$H frequency of 600 MHz using a Bruker Avance spectrometer equipped with a BBI probe. All experiments were conducted at RT, 25°C. 3 mm diameter NMR tubes with a sample volume of 200 µL in all experiments. Solutions were buffered using an $H_2O$ PBS buffer corrected to pH 7.4. The sample preparation is exemplified as follows; the compound (10 µL of a 10 mM solution in DMSO-$d_6$) was added to an Eppendorf tube before sequential addition of the $H_2O$ PBS buffer (163.6 µL), $D_2O$ (20 µL), and protein (6.4 µL, 311.8 µM). The resulting solution was vortexed to mix and transferred to a 3 mm NMR tube prior to the NMR analysis.

For competition experiments using anti-RAS scFv, protein preparation for NMR was carried out in a similar manner; the compound (10 µL of a 10 nM solution in DMSO-$d_6$) was added to an Eppendorf tube before sequential addition of the $H_2O$ PBS buffer (146.4 µL), $D_2O$ (20 µL), protein (6.4 µL, 311.8 µM) and anti-RAS scFv (17.2 µL, 116.6 µM). The resulting solution was vortexed to mix and transferred to a 3 mm NMR tube prior to the NMR analysis.

Negative controls (compound alone) were prepared in a similar manner, in order to obtain an end volume of 200 µL.

## Chemical synthesis procedures

All reactions involving moisture-sensitive reagents were carried out under a nitrogen atmosphere using standard vacuum line techniques and glassware that was flame-dried before use. Anhydrous solvents were prepared following the procedure outlined (*Pangborn et al., 1996*). Water was purified by an Elix UV-10 system. All other solvents and reagents were used as supplied (analytical or HPLC grade) without prior purification. Brine refers to a sat. aq. solution of NaCl. *In vacuo* refers to the removal of solvent by the use of a rotary evaporator attached to a diaphragm pump.

Thin layer chromatography was performed on normal phase Merck silical gel 60 F254 aluminum-supported thin layer chromatography sheets. Visualization of spots was either by absorption of ultra violet light (λmax 254 nm), or by thermal development after staining with 1% aq. KMnO4. Flash column chromatography was performed on Kieselgel 60 silica in a glass column, under a positive pressure.

NMR spectra were recorded on Bruker Avance spectrometer (AVIII 600) in the deuterated solvent stated. The field was locked by external referencing to the relevant deuteron resonance. Chemical shifts (δ) are reported in parts per million (ppm). The multiplicity of each signal is indicated by: app. (apparent), s (singlet), br s (broad singlet), d (doublet), t (triplet), q (quartet), dd (doublet of doublets) or m (multiplet). Coupling constants (*J*) are quoted in Hz and are reported to the nearest 0.1 Hz.

Low-resolution mass spectra were recorded on an Agilent 6120 spectrometer operating in positive or negative mode, from solutions of MeOH. Accurate mass measurements were run on either a Bruker MicroTOF internally calibrated with polyalanine, or a Micromass GCT instrument fitted with a

Scientific Glass Instruments BPX5 column (15 m x 0.25 mm) using amyl acetate as a lock mass, by the mass spectrometry department of the Chemistry Research Laboratory, University of Oxford, UK. m/z values are reported in Daltons.

## 5-bromo-2,3-dihydrobenzo[b][1,4]dioxine (3)

**1**          **2**          **3**

**Chemical structure 1.**
DOI: https://doi.org/10.7554/eLife.37122.017

A solution of 2-bromo-6-methoxyphenol **1** (2.50 g, 12.3 mmol) in $CH_2Cl_2$ (80 mL) was cooled to −78°C before dropwise addition of $BBr_3$ (1 M in heptane, 14.8 mL, 14.8 mmol). The resulting mixture was warmed to room temperature and stirred for 2 hr before being poured onto an ice/water (200 mL) and stirred for 30 min. The organic phase was separated, washed with water (100 mL) and brine (100 mL), dried ($Na_2SO_4$), filtered and concentrated *in vacuo* to give the desired 3-bromobenzene-1,2-diol two as a brown oil (2.24 g, 11.9 mmol, 97%), which was used in the next step without further purification.

A solution of diol **2** (1.00 g, 5.35 mmol) in DMF (20 mL) was treated sequentially with $K_2CO_3$ (1.77 g, 12.8 mmol), and 1,2-dibromoethane (507 μL, 5.88 mmol) before being heated to 60°C for 18 hr. The reaction was then cooled down before addition of water and brine (1:1, 50 mL) and EtOAc (100 mL). The organic phase was washed further with water and brine (1:1, 4 × 50 mL), dried ($Na_2SO_4$), filtered and concentrated *in vacuo* to give the crude material as a brown oil. Purification on silica gel (pentane/EtOAc, 4:1) afforded the desired 5-bromo-2,3-dihydrobenzo[b][1,4]dioxine three as a clear oil (1.11 g, 5.19 mmol, 97%).

## 5-(4-chloro-3-methoxyphenyl)−2,3-dihydrobenzo[b][1,4]dioxine (4)

**3**          **4**

**Chemical structure 2.**
DOI: https://doi.org/10.7554/eLife.37122.018

Bromide **3** (600 mg, 2.79 mmol) was added to a vial before addition of 1,4-dioxane/water (5:1, 8 mL); the solution was degassed before sequential addition of $K_2CO_3$ (1.16 g, 8.37 mmol), 4-chloro-3-methoxyphenyl boronic acid (572 mg, 3.07 mmol), and Pd(dppf)Cl$_2$ (100 mg, 0.140 mmol). The vial was sealed and the reaction heated to 100°C for 18 hr, cooled down and concentrated *in vacuo*. The residue was purified on silica gel (pentane/EtOAc, 9:1) to afford the desired 5-(4-chloro-3-methoxyphenyl)−2,3-dihydrobenzo[b][1,4]dioxine four as a clear oil (745 mg, 2.70 mmol, 97%). [1]H NMR (600 MHz, CDCl$_3$) δ 7.39 (1H, d, *J* 8.1 Hz), 7.11 (1H, s), 7.08 (1H, dd, *J* 8.2, 1.7 Hz), 6.91–6.89 (3H, m), 4.31–4.28 (4 hr, m), 3.94 (3H, s); [13]C NMR (150 MHz, CDCl$_3$) δ 154.5, 143.9, 140.6, 137.5,

130.0, 129.6, 122.6, 122.4, 121.4, 121.1, 117.0, 113.5, 64.4, 64.1, 56.2; $m/z$ (ESI$^+$) 277 ([M + H]$^+$); HRMS (ESI$^+$) [C$_{15}$H$_{14}$ClO$_3$] requires 277.0631, found 277.0591.

## 4-(2,3-dihydrobenzo[b][1,4]dioxin-5-yl)-N-(4-(dimethylamino)methyl)phenyl)−2-methoxyaniline (3344)

**Chemical structure 3.**
DOI: https://doi.org/10.7554/eLife.37122.019

Chloride **4** (75 mg, 0.272 mmol), Cs$_2$CO$_3$ (266 mg, 0.866 mmol), 3-((dimethylamino)methyl)aniline (61 mg, 0.408 mmol), XPhos (13 mg, 0.027 mmol) and Pd(OAc)$_2$ (3 mg, 0.014 mmol) were added sequentially to a vial and degassed with N$_2$ for 5 min. Degassed 1,4-dioxane (2 mL) was then added, the vial sealed and heated to 100°C for 18 hr. The mixture was cooled down, diluted with EtOAc (30 mL), and washed with a 50/50 solution of water and brine (2 × 30 mL). The organic phase was dried (Na$_2$SO$_4$) and concentrated *in vacuo*. Purification by column chromatography on silica gel (CH$_2$Cl$_2$/MeOH, 9:1) afforded the desired 4-(2,3-dihydrobenzo[b][1,4]dioxin-5-yl)-N-(3-((dimethylamino)methyl)phenyl)−2-methoxyaniline **3344** as a yellow oil (102 mg, 96%).

$^1$H NMR (400 MHz, MeOD) δ 7.26 (1H, d, *J* 8.3 Hz), 7.20 (1H, dd, *J* 7.6, 0.2 Hz) 7.12 (1H, d, *J* 2.0 Hz) 7.08–7.04 (2H, m), 7.00 (1H, dd, *J* 8.3, 2.0 Hz), 6.88 (1H, dd, *J* 7.6, 2 Hz), 6.83 (2H, *J* 7.8, 0.2 Hz), 6.78 (1H, dd, *J* 7.8, 2.0 Hz), 4.25–4.20 (4H, m), 3.87 (3H, s), 3.45 (2H, s), 2.27 (6H, s), NH was not observed; $^{13}$C NMR (125 MHz, CDCl$_3$) δ 150.2, 145.5, 145.3, 142.2, 139.4, 133.2, 132.4, 131.7, 130.3, 123.5, 123.0, 122.9, 122.0, 120.1, 118.3, 117.1, 116.7, 113.6, 65.8, 65.5, 65.1, 56.4, 45.3; $m/z$ (ESI$^-$) 38 ([M–H]$^-$); HRMS (ESI$^-$) [C$_{24}$H$_{25}$N$_2$O$_3$] requires 389.1865, found 389.1841.

## $^1$H CPMG NMR experiments for compound Kd calculation

Typical experimental parameters for Carr-Purcell-Meiboom-Gill (CPMG) NMR spectroscopy were the following: total echo time, 40 ms; relaxation delay, 2 s; and number of transients, 264 (Abboud et al., 2016). The PROJECT-CPMG sequence (90°x-[T−180°y-T- 90°y-T−180°y-T]$_n$-acq) was applied. Water suppression was achieved by presaturation. Prior to Fourier transformation, the data were multiplied with an exponential function with 3 Hz line broadening. The CPMG experiments were conducted at a $^1$H frequency of 700 MHz using a Bruker Avance with 5 mm inverse TCI 1 hr/13C/15N cryoprobe. All experiments were conducted at RT and lapsed 128 scans. 3 mm diameter NMR tubes with a sample volume of 200 μL were used in all experiments. Solutions were buffered using a D$_2$O PBS buffer corrected to pH 7.4. The sample preparation is exemplified as follows: for a 5 μM GST-KRAS$^{G12V}$ sample: 55 μM of the 3344 compound (1.1 μL of a 10 mM solution in DMSO-$d_6$) was added to an Eppendorf before sequential addition of the D$_2$O PBS buffer (194.0 μL) and GST-KRAS$^{G12V}$ (4.9 μL of a 205 μM solution, the protein is in an H$_2$O buffer for stability reason). The resulting solution was vortexed to be fully mixed and transferred to a 3 mm NMR tube before the run. Negative controls (compound alone, without the KRAS protein) were prepared in a similar manner, in order to obtain an end volume of 200 μL.

CPMG experiments were carried out at a fixed 3344 concentration (55 μM, optimal concentration for these CPMG NMR experiments) and a variable GST-KRAS$^{G12V}$ concentration. The amount of GST-KRAS$^{G12V}$ was increased from 0 μM until the signals of the compound completely disappear in the proton NMR at 20 μM. Seven measurements were done in total with 0 μM, 2.5 μM, 5 μM, 7.5

µM, 10 µM, 15 µM and 20 µM of GST-KRAS$^{G12V}$. The integrations of the protons acquired were all compared to the compound alone (with no KRAS) in order to obtain a percentage decrease for each concentration of KRAS. Three different proton signals were used and a mean was calculated for each run. KRAS concentration experiments were run in triplicate and a mean was also calculated for each concentration. Concentration and percentage of decrease were plotted and Kd fitting was run on the generated curve using Origin 2017 software with the following function: A*(1/ (2*C))* ((B + x + C)-sqrt(((B + x + C)2)-(4*x*C))) where A is the maximum % of inhibition (*i.e.* 100), B is the Kd, C is the concentration of compound and x the concentration of KRAS protein necessary to reach 100% of signal reduction of the compound.

## Recombinant protein expression for crystallography and NMR: KRAS$^{G12V}$, KRAS$^{Q61H}$ and scFv

KRAS$^{G12V}$ cDNA was cloned into the pGEX vector in-frame with an N-terminal Glutathione-S trans-ferase (GST) tag. pGEX-GST-KRAS$^{G12V}$ was transformed into *E.coli* BL21 (DE3) cells. Bacterial cells were cultured at 37°C to an OD$_{600}$ of 0.5 and induced with IPTG (isopropyl 1-thio-beta-D-galactopyr-anoside, final concentration 0.1 mM) at 16°C overnight. The bacteria cultures were harvested by cen-trifugation and the cell pellets re-suspended in 50 mM Tris-HCl pH8.0, 140 mM NaCl, 1 mM mercaptoethanol supplemented with complete protease inhibitor (Roche). The GST-fusion proteins were purified by glutathione-sepharose column chromatography (GE Healthcare) and eluted with 50 mM Tris-HCl pH8.0, 10 mM reduced glutathione, 1 mM mercaptoethanol, 5 mM MgCl$_2$.

KRAS$^{Q61H}$ cDNA was cloned into the pRK-172 vector in-frame with an N-terminal 6xHis-tag and TEV protease recognition site. The plasmid containing KRAS$^{Q61H}$ sequence was transformed into *E. coli* B834(DE3) pLysS cells, which were grown in 25 mL LB medium with 50 µg/mL Carbenicillin and 34 µg/mL Chloramphenicol for 16 hr, prior to inoculation of 1L LB medium. Protein expression was induced at OD$_{600}$ = 0.6 by addition of IPTG to a final concentration of 0.5 mM and cells grown over-night at 16°C. Bacteria were harvested by centrifugation and sonicated in 50 mM Tris-HCl, pH 7.5, 500 mM NaCl, 5 mM MgCl$_2$ and 10 mM imidazole and EDTA-free protease inhibitor cocktail (Roche Diagnostics). Proteins were purified using nickel agarose beads (Invitrogen) and bound proteins were eluted batch-wise in 50 mM Tris-HCl, pH 7.5, 500 mM NaCl, 5 mM MgCl$_2$ and 300 mM imidaz-ole. RAS protein samples were concentrated using Vivapore 10/20 mL concentrator (7.5 kDa molecu-lar weight cut-off; Sartorius Vivapore) to a final volume of approximately 1 mL. Nucleotide exchange for crystallographic samples was carried out following published procedures (*Herrmann et al., 1996*). RAS proteins were further purified by gel filtration on a HiLoad Superdex 75 10/300 GL col-umn (GE Healthcare) in a buffer containing 20 mM HEPES pH 8.0, 150 mM NaCl, 5 mM MgCl$_2$ and 1 mM DTT at a flow rate of 0.5 mL/min. Fractions corresponding to the protein were pooled and con-centrated to 45–75 mg/mL for crystallization trials. Protein concentration was determined by extinc-tion coefficient (ε$_{280}$ = 12045 L/mol/cm). Protein purity was analyzed by SDS-PAGE stained with Coomassie Brilliant Blue. scFv recombinant protein was expressed and purified as described else-where (*Tanaka et al., 2007*).

## Crystal structure and 3344 soaking

For X-ray diffraction experiments, KRAS$^{Q61H}$-GppNHp crystals were grown by vapour diffusion at 4°C by mixing 1.5 + 1.5 volumes of KRAS solution at a concentration of 75 mg/mL KRAS$^{Q61H}$, with 8–15% w/v Polyethylene Glycol 3350 and 0.2 M lithium citrate pH 5.5. The resulting crystals are termed crystal form I hereafter. Prior to X-ray data collection, crystals were cryo-protected by addi-tion of 20% glycerol to the crystallization buffer and flash-cooled in liquid nitrogen. 3344 was initially dissolved at 200 mM in 100% DMSO and sequentially mixed in a ratio of 1:1 with crystallization buffer (8–15% w/v Polyethylene Glycol 3350, 0.2 M lithium citrate 7.0 and 20 mM Tris-HCl pH 7.0) to give a final concentration of compound of 50 mM and 25% DMSO in a 5 µL drop. Soaked crystals were flash-cooled in liquid nitrogen prior to data collection using the final DMSO concentration on the soaking drop as cryo-protectant. X-ray diffraction data were collected at beamline ID30A-1 (*Bowler et al., 2015*; *Bowler et al., 2016*; *Nurizzo et al., 2016*; *Svensson et al., 2015*) at The Euro-pean Synchrotron Radiation Facility (ESRF, Grenoble, France). The structure of KRAS$^{Q61H}$ GppNHp-3344 was solved by molecular replacement using a KRAS169$^{Q61H}$ GPPNHP-Abd-2, (PDB ID 5OCO) as a search model within the program Phaser (*McCoy, 2007*; *McCoy et al., 2007*). Structures were

manually adjusted using COOT (*Emsley et al., 2010*) and refined using REFMAC (*Murshudov et al., 1997*). Crystal Form I (KRAS$^{Q61H}$) has six KRAS molecules in the asymmetric unit, assembled as a hexamer. Electron density maps averaged with six-fold non-crystallographic symmetry (NCS) were used to improve the definition of the bound compounds. Refinements were also performed with the six fold NCS applied. The refined models were validated using PROCHECK (*Laskowski et al., 1993a*), MolProbity (*Chen et al., 2010*) and Phenix software packages (*Adams et al., 2010*; *Laskowski et al., 1993b*). Figures were created using PyMOL (Schrodinger). Data collection and refinement statistics are summarized in *Table 1*.

## Quantification and statistical analysis

All quantifications were performed using ImageJ or Prism 7.0 c (GraphPad Software), BRET titration curves and statistical analysis were performed using Prism 7.0 c (GraphPad Software). Data are typically presented as mean ± SD or SEM as specified in the figure legends. Statistical analyses were performed with a one-way ANOVA followed by Dunnett's post-hoc tests or Sidak's post-hoc tests unless otherwise indicated in the figure legends. *$p<0.05$, **$p<0.01$, ***$p<0.001$, ****$p<0.0001$.

## Data and software availability

Structure files and coordinates have been deposited to PDB under this accession number: 6F76.

# Acknowledgements

This work was funded by grants from Wellcome Trust 100842/Z/12/Z, 099246/Z/12/Z, Bloodwise 12051 and MRC MR/J000612/1. We would like to thank Drs Roger Williams and Olga Perisic for the PI3K and p85 plasmids, Jennifer Chambers for KRAS$^{G12D\ FL}$ plasmid and Dr James Felce for the GFP$^2$ and RLuc8-N3 plasmids. We acknowledge the European Synchrotron Radiation Facility for provision of synchrotron radiation facilities and we would like to thank Matthew Bowler for assistance in using beamline ID30A-1. We also thank Drs. Lydia Lee, Phillip Fallon, Jonathan Dunn, Robert Freem, Traore Tenin and Sophie Williams from Domainex for chemistry. We also acknowledge Professor Tim Claridge for his help in the setup of the waterLOGSY and CPMG NMR experiments and Dr Daniel Ebner for access to Envision 2103 plate reader.

# Additional information

### Competing interests

Abimael Cruz-Migoni: Employed by Immunocore; no other competing financial interests to declare. Angela Russell: Founder of OxStem; no other competing financial interests to declare. The other authors declare that no competing interests exist.

### Funding

| Funder | Grant reference number | Author |
| --- | --- | --- |
| Medical Research Council | MR/J000612/1 | Terence H Rabbitts |
| Wellcome | 099246/Z/12/Z | Terence H Rabbitts |
| Bloodwise | 12051 | Terence H Rabbitts |
| Wellcome | 100842/Z/12/Z | Terence H Rabbitts |

The funders had no role in study design, data collection and interpretation, or the decision to submit the work for publication.

### Author contributions

Nicolas Bery, Conceptualization, Methodology, Formal analysis, Investigation, Writing—original draft, Writing—review and editing; Abimael Cruz-Migoni, Formal analysis, Investigation, Writing—original draft; Carole JR Bataille, Hanna Tulmin, Formal analysis, Writing—original draft; Camilo E Quevedo, Formal analysis, Writing—original draft, Writing—review and editing; Ami Miller,

Investigation, Writing—review and editing; Angela Russell, Formal analysis, Supervision; Simon EV Phillips, Stephen B Carr, Formal analysis, Supervision, Writing—original draft, Writing—review and editing; Terence H Rabbitts, Conceptualization, Formal analysis, Supervision, Funding acquisition, Investigation, Methodology, Writing—original draft, Project administration, Writing—review and editing

## Author ORCIDs
Nicolas Bery http://orcid.org/0000-0002-2643-3897
Terence H Rabbitts http://orcid.org/0000-0002-4982-2609

## Decision letter and Author response
Decision letter https://doi.org/10.7554/eLife.37122.025
Author response https://doi.org/10.7554/eLife.37122.026

# Additional files

## Supplementary files
• Supplementary file 1. DNA and protein sequences of BRET biosensors constructs. The list of the DNA and protein sequences from the different RAS BRET biosensor constructs used in this study.
DOI: https://doi.org/10.7554/eLife.37122.020
• Transparent reporting form
DOI: https://doi.org/10.7554/eLife.37122.021

## Data availability
Diffraction data have been deposited in PDB ID 6F76.

The following dataset was generated:

| Author(s) | Year | Dataset title | Dataset URL | Database, license, and accessibility information |
|---|---|---|---|---|
| Bery N, Cruz-Migoni A, Quevedo CE, Phillips SVE, Carr S, Rabbitts TH | 2018 | Antibody derived (Abd-8) small molecule binding to KRAS. | http://www.rcsb.org/pdb/results/results.do?tabtoshow=Unreleased&qrid=60BEFAF0 | Publicly available at the RCSB Protein Data Bank (accession no. 6F76) |

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
