## [Decision Letter]

Thank you for submitting your article "BRET-based RAS biosensors that show a novel small molecule is an inhibitor of RAS-effector protein-protein interactions" for consideration by *eLife*. Your article has been reviewed by four peer reviewers, including Roger Davis as the Reviewing Editor, and the evaluation has been overseen Philip Cole as the Senior Editor. The following individual involved in review of your submission has agreed to reveal his identity: David Lambright (Reviewer #3).

The reviewers have discussed the reviews with one another and the Reviewing Editor has drafted this decision to help you prepare a revised submission.

Summary:

This is a resource paper that describes the design, construction, and use of a series of BRET-based RAS biosensors that can be used to characterize inhibitors of RAS-effector protein-protein interactions. This approach will be of great interest to the cancer research community. The high quality data presented support the conclusions of the study. However, there are a number of points that require clarification.

Essential revisions:

1) The Kd identified for the 3344 compound binding to KRAS is not completely clear. The titration study (Figure 1—figure supplement 2D) is interpreted to show an approximate Kd of 126nM, but the x-axis shows that the lowest concentration of 3344 studied is 2.5µM, which is many-fold above the estimated Kd and would presumably saturate the interaction. The interpretation of this study is therefore unclear. Moreover, while the x-axis is labeled as the concentration of 3344, the figure legend suggests that the x-axis refers to GST-KRAS. Please explain. The authors should consider the use of an alternative biophysical technique to confirm the Kd.

2) These types of probes are usually subject to multiple rounds of optimization. However, no information is provided concerning the optimization procedures that were employed. The absence of this information means that it is unclear whether the BRET probes described can be further optimized.

3) The DNA constructs are described in outline form. As a resource paper, it would be very helpful to the reader if more information could be provided. For example, supplementary files showing full sequence information should be included.

4) No information is provided concerning how the small molecule 3344 was designed/identified.

5) The structure of the control small molecule Abd-2 is not presented. Why was this molecule chosen rather than any other control molecule? How was this molecule obtained?

6) The amount of bleed-through between the two channels should be characterized, and corrections for this bleed-through should be included in the calculations. RAS-RLuc8 alone and GFP alone should be used to characterize such bleed-through.

7) The authors should comment on and address the amount of autoluminescence and autofluorescence from untransfected cells.

8) Long incubation times (20h) are employed in the inhibitor studies of Ras signaling (e.g. Figures 3 and 4). This is a potential problem for studies to dissect signaling mechanisms because the inhibitor may act on signaling pathways indirectly by altering autocrine pathways (e.g. by blocking cytokine expression) that cause secondary activation of the ERK and AKT pathways. Short-term inhibition assays would prevent this caveat on the interpretation of these data.

9) Figure 6B is simply an enlargement that does not provide additional information beyond what is obvious in Figure 6A. This panel should be replaced by a panel illustrating how the compound interferes with effector binding – this would support conclusions concerning the mechanism of inhibition.

10) Although 3344 does not interact with the switch regions in the crystal structure, it would be helpful to clarify whether there are any changes (or not) in the conformation or b-factors in the switch regions compared with the un-soaked crystal structure.

---

## [Author Response]

Essential revisions:1) The Kd identified for the 3344 compound binding to KRAS is not completely clear. The titration study (Figure 1—figure supplement 2D) is interpreted to show an approximate Kd of 126nM, but the x-axis shows that the lowest concentration of 3344 studied is 2.5µM, which is many-fold above the estimated Kd and would presumably saturate the interaction. The interpretation of this study is therefore unclear. Moreover, while the x-axis is labeled as the concentration of 3344, the figure legend suggests that the x-axis refers to GST-KRAS. Please explain. The authors should consider the use of an alternative biophysical technique to confirm the Kd.

This figure unfortunately has an error and for clarification the x-axis should be KRAS concentration and not 3344 concentration (we have amended the figure accordingly). We have also added additional explanations in the corresponding figure legend and Materials and methods section to clarify how the 3344 Kd was estimated with the ^1^H CPMG NMR method. Regarding the Kd measurement, the focus of our Resource paper the implementation of the complete set of BRET2 reagents and using the 3344 compound as an exemplar. We have used the saturation CPMG NMR as a method to assay Kd since this is considered a reliable approach (e.g. Baldwin and Kay (2009); Abboud et al. (2016)) and utilises technology and expertise available locally. The detailed characterisation of 3344 would be the subject of a larger medicinal chemistry study that we hope to undertake later when resources allow.

2) These types of probes are usually subject to multiple rounds of optimization. However, no information is provided concerning the optimization procedures that were employed. The absence of this information means that it is unclear whether the BRET probes described can be further optimized.

As the referees suggest, we indeed performed optimization steps on our biosensors but did not included the data in the study in a first instance. Different parameters need to be controlled to get an optimal BRET signal such as the fusion position of the RLuc8 and GFP^2^ moieties on their respective protein. Indeed, even if two proteins interact the GFP^2^ and the RLuc8 moieties need to be close enough (<10 nm) to induce a detectable and reliable BRET signal. Therefore, our BRET probes were optimized in the following ways. For our study, many RAS-effectors crystal structures are already available allowing us to rationalize the design of the donor and acceptor pairs with respect to the spatial arrangement of the donor and acceptor part of the BRET pairs. As the RLuc8 molecule had to be on the N-terminus end of RAS protein, since we used full-length RAS protein (with the prenylation site), we checked which position (N or C-terminus) on the effectors, or iDAbs, the GFP^2^ moiety should be for proximity of the RLuc8 moiety to the donor. If the structure was not available, we used a try and test strategy (e.g. with CRAF^FL^ and PI3Kα^FL^). We have added these data in the new Figure 2—figure supplement 1A and new Figure 4—figure supplement 1A.

We also optimized the linker length between the donor/acceptor moieties and the protein of interest. Three different linker lengths have been tested for two BRET pairs: KRAS^G12D^/CRAF RBD and KRAS^G12D^/iDAb RAS. Accordingly, we applied the same linker length (GGGS)_3_ for all the effector constructs and (GGGS)_2_ for the iDAb constructs. The RAS linker was always (GGGS)_3_ as very little difference was visible on the BRET ratio between the three linker tested. An additional supplementary figure has been added showing these linker optimization data (new Figure 1—figure supplement 1A). We also added a discussion of the optimization in the Results section.

3) The DNA constructs are described in outline form. As a resource paper, it would be very helpful to the reader if more information could be provided. For example, Supplementary files showing full sequence information should be included.

This sequence information has now been included for all the constructs used in our study (Supplementary file 1).

4) No information is provided concerning how the small molecule 3344 was designed/identified.

The 3344 molecule was one of a set of compounds that arose from a RAS/antibody-fragment based medicinal chemistry programme. It was chosen to validate our BRET2 biosensor toolbox because its binding to RAS was inhibited by the anti-RAS intracellular scFv. Furthermore, we had found full electron density of the compound in KRAS crystal soaking work that validates the use of that compound. An explanatory statement about 3344 has been added to the main text.

5) The structure of the control small molecule Abd-2 is not presented. Why was this molecule chosen rather than any other control molecule? How was this molecule obtained?

The structure of Abd-2 has been added in the Figure 1—figure supplement 3F. This molecule originated from an SPR screen of a fragment library using HRAS and, because it was closely related to an initial hit, it has very low affinity for RAS binding and therefore serves a good and relevant control compound. An explanatory statement has been added to the main text.

6) The amount of bleed-through between the two channels should be characterized, and corrections for this bleed-through should be included in the calculations. RAS-RLuc8 alone and GFP alone should be used to characterize such bleed-through.

All the BRET measurements from our study are background corrected with the donor only construct as specified in the corresponding Materials and methods section. Indeed, GFP^2^ constructs alone do not give any background when tested with the BRET substrate and are usually not taken into account in the BRET background correction (Pfleger et al., 2006) (new Figure 1—figure supplement 1B). We added an explanatory statement to the Results section.

7) The authors should comment on and address the amount of autoluminescence and autofluorescence from untransfected cells.

The autoluminescence of untransfected cells is negligible compared to the RLuc8 transfected cells, which is why it was not subtracted from the BRET calculation. The autofluorescence of untransfected cells is similar to the GFP^2^ signal detected with cells transfected with the RLuc8 construct only (new Figure 1—figure supplement 1B). We have commented about this in the Results section.

8) Long incubation times (20h) are employed in the inhibitor studies of Ras signaling (e.g. Figures 3 and 4). This is a potential problem for studies to dissect signaling mechanisms because the inhibitor may act on signaling pathways indirectly by altering autocrine pathways (e.g. by blocking cytokine expression) that cause secondary activation of the ERK and AKT pathways. Short-term inhibition assays would prevent this caveat on the interpretation of these data.

We agree that short-term assays are important to avoid potential misinterpretation/ indirect effect of the compound on RAS signalling data. Figures 6C and D did show a short-term inhibition (3h incubation rather than 20h). However, we performed further short-term incubation of the inhibitor (3 hours incubation) in BRET and Western Blot experiments and present these new data in the new Figure 2—figure supplement 1E-G, new Figure 3—figure supplement 1D-F and new Figure 4—figure supplement 1D-F.

We observed that, after 3 hours of incubation with the compound, there was a decrease of the interaction of RAS with CRAF^FL^ and PI3Kα^FL^ (BRET experiments), which correlated with a decrease of pMEK, pERK and pAKT signal (Western blot experiments) showing the direct effect of the compound on RAS signalling. We also added a commentary in the corresponding Results section.

9) Figure 6B is simply an enlargement that does not provide additional information beyond what is obvious in Figure 6A. This panel should be replaced by a panel illustrating how the compound interferes with effector binding – this would support conclusions concerning the mechanism of inhibition.

As suggested, Figure 6B has been replaced by the superimposition of the structures of three RAS-effector protein complexes with the structure of KRAS-3344 complex. These data show an overlap between the compound and the bound effectors. Therefore, the competition effect of 3344 can be explained by straightforward steric hindrance.

10) Although 3344 does not interact with the switch regions in the crystal structure, it would be helpful to clarify whether there are any changes (or not) in the conformation or b-factors in the switch regions compared with the un-soaked crystal structure.

We have not directly compared the KRAS crystals alongside those soaked with 3344 in one experiment. However, we do not detect conformation alterations by aligning superimposition of KRAS with and without 3344.